# Stress granule phase separation in stress-responsive cytosolic extract-in-oil droplets

Aline Lieber [1], Oskar Staufer [2,3], Zhaozhi Sun[1,8], Ulrike Engel [4], Charlotte Flory [5], Nathan Mikhaylenko[3], Kevin Jahnke [2,9], Katja Kopp [1,10], Philipp Klein[1,11], Sarah Hofmann [6,12], Oliver T. Fackler [7], Pavel Ivanov [6], Ilia Platzman[2], Pietro Scaturro [5], Joachim P. Spatz[2] & Alessia Ruggieri [1] ✉

Stress granules (SGs) are biomolecular condensates that form transiently in the cytosol of mammalian cells in response to stress. Dysregulation of their assembly or disassembly is implicated in human age-related diseases. While phase separation is the key process underlying SG assembly, understanding of their function, composition and regulation in response to physiological stimuli is limited. This knowledge gap reflects the challenge of gaining comprehensive and quantitative insights into the dynamic regulation of the complex composition of SGs at the single-cell level. Here we present an emulsion-based microfluidics method to overcome this limitation. "Cytosolic extracts-in-oil droplets" (CEODs) recreate a confined active cytosolic milieu that undergoes phase separation and SG formation in response to stress under physiological conditions. This approach led to the discovery of seven previously unrecognised SG components involved in signalling pathways. CEODs provide a versatile and cost-effective screening platform for future mechanistic and therapeutic studies.

Stress granules (SGs) are biomolecular condensates that transiently form in the cytosol when cells are exposed to potentially hazardous conditions. Dysregulation of pathways leading to SG assembly or disassembly is increasingly associated with human diseases related to ageing, such as neurodegeneration, underscoring their central physiological roles[1–3]. These membraneless subcellular assemblies are formed as a result of two distinct pathways leading to translation initiation repression. The first occurs in response to activation of the integrated stress response (ISR), the main pathway controlling protein synthesis. Upon cellular imbalances, one or several of the four mammalian stress kinases—HRI, GCN2, PERK and PKR—phosphorylate the alpha subunit of the eukaryotic initiation factor 2 (eIF2α), causing translation initiation stalling and polysome disassembly[4]. The second pathway occurs upon interference with eIF4F assembly at the 5′ end of messenger RNAs (mRNAs, e.g., by drug inhibiting the cap complex helicase eIF4A[5–7].

[1]Department of Infectious Diseases, Molecular Virology, Center for Integrative Infectious Disease Research (CIID), Heidelberg University, Medical Faculty Heidelberg, Heidelberg, Germany. [2]Department for Cellular Biophysics, Max Planck Institute for Medical Research, Heidelberg, Germany; Institute for Molecular Systems Engineering (IMSE), Heidelberg University, Heidelberg, Germany; Max Planck-Bristol Centre for Minimal Biology, University of Bristol, Bristol, UK. [3]INM - Leibniz Institute for New Materials, Saarbrücken, Germany. [4]Nikon Imaging Center at Heidelberg University and Centre for Organismal Studies (COS), Heidelberg University, Heidelberg, Germany. [5]Leibniz Institute of Virology, Hamburg, Germany. [6]Brigham and Women's Hospital, Harvard Medical School, Harvard Initiative for RNA Medicine, Boston, MA, USA. [7]Department of Infectious Diseases, Integrative Virology, Center for Integrative Infectious Disease Research (CIID), Heidelberg University, Medical Faculty Heidelberg, Heidelberg, Germany. [8]Present address: Shenzhen Bay Laboratory, Gaoke International Innovation Center, Guangdong Province, Shenzhen, China. [9]Present address: Biomembrane Engineering, Max Planck Institute for Medical Research, Heilbronn, Germany. [10]Present address: BioSpring GmbH, Frankfurt, Germany. [11]Present address: GSK Vaccines, Rixensart, Belgium. [12]Present address: Anatomy and Developmental Biology Heidelberg University, Medical Faculty Mannheim, Mannheim, Germany. ✉e-mail: alessia.ruggieri@med.uni-heidelberg.de

Biomolecular condensates have emerged as important dynamic regulators of biochemical reactions[8–10]. They organise according to the common principle of liquid-liquid phase separation (LLPS), which involves the spatiotemporal control of a collection of macromolecules (such as proteins) undergoing condensation and dissolution[11]. The process is reversible and governed by weak, non-covalent, and multivalent homotypic and heterotypic interactions between proteins and nucleic acids, and facilitated by proteins harbouring defined RNA recognition motifs and intrinsically disordered regions (IDRs)[12]. Thus, under conditions of stress and mRNA translation repression, the subsequent increase in the local concentration of untranslated messenger ribonucleoproteins (mRNPs) facilitates their phase separation with other RNA-binding proteins (RBPs) and the formation of SGs[13–16].

Ras GTPase-activating protein-binding proteins 1 and 2 (G3BPs) are two indispensable, central drivers of SG condensation[17–19]. Their binding to RNAs is enhanced upon increased availability of untranslated mRNA, promoting dimerisation and cooperative interactions with a network of SG core proteins[20,21]. Amongst these, Caprin1, TIA1, FMR1 and DDX3 contribute to SG nucleation and growth with varying valencies[22]. G3BPs are therefore regarded as "scaffold" or "nucleating" proteins essential for SG condensation, as opposed to "client" proteins whose depletion has no or little impact on SG formation[23]. Increasing evidence supports the existence of an interaction networks between G3BPs, other SG components and mRNAs also in the absence of stress. This network of mRNPs, so-called "seeds" or "cores", would remodel upon exposure to stress due to the increase in local concentration of untranslated mRNAs, resulting in the condensation of a larger microscopically visible SG[24–26]. At the ultrastructural level, SGs exhibit spatially discrete substructures[24,27–30] where components have different dynamic behaviours[15,29,31–34]. These observations are consistent with a "core-shell" model in which molecule diffusion is more limited in the core region of SGs, while increased in the more dilute surrounding phase in equilibrium with the cytosol[24,26].

The precise regulation of dynamic multivalent interactions between SG components is likely to influence SG reversibility and compositional specificities but remains poorly understood. The RNA and protein composition of SGs varies according to the type of stress experienced, between different cell types, over time, and in the context of disease[35–41]. SGs have been proposed to serve as sites for the spatiotemporal control of molecules or of their activity, e.g., biochemical modifications, sequestration, and protection from degradation[42,43]. This may account for the diversity of functions proposed for SGs. For instance, they have been implicated in cell survival and death mechanisms, viral infections, and cancer[44–47]. Moreover, pathological persistent SGs have been associated with ageing and neurodegenerative diseases[1,2]. Although attaining a fundamental characterisation of the biophysical properties of SGs is a compelling goal, little understanding has yet been achieved, mainly due to their extensive complexity.

The implementation of bottom-up approaches, which entail the simplification of complex processes to their most fundamental units, has been particularly useful and significantly advanced our biophysical and biochemical comprehension of SG formation, revealing the pivotal role of phase separation in their condensation[12,48–50]. Bottom-up approaches provide valuable information about the fundamental properties and capacities of the minimal components that form SGs, but do not reflect their actual occurrence in a much more complex environment. Indeed, the in vitro condensation of purified recombinant SG proteins is governed by a number of non-physiological parameters. These include high protein or salt concentrations, the addition of nucleic acids, or the addition of macromolecular crowding agents such as polyethylene glycol (PEG), which facilitates LLPS through volume exclusion[51].

Recent methods for SG reconstitution based on cellular extracts have provided an elegant way of narrowing the gap between bottom-up and cell-based studies[52,53]. While these systems offer valuable insights, they do not allow the study of physiological responses of cells to stress. Stress-induced signalling cascades regulate post-translational modifications (PTMs) and influence the assembly and disassembly of SGs[43,54], which likely has implications for their composition and function, particularly in a pathological context. To overcome this limitation, we have devised a methodology for studying and manipulating SG phase separation in vitro, which offers a more sophisticated level of complexity. We employed emulsion-based droplet microfluidics to recreate an in vitro environment that resembles the confined active cytosolic milieu. Such "cytosolic extracts-in-oil droplets" (CEODs) undergo phase separation and SG condensation in response to stress under physiological conditions. They are scalable, cost-effective and can be used for other downstream applications, such as high-throughput and candidate-based in vitro screenings. Moreover, using this innovative, unbiased stress-responsive methodology, we identified new SG client proteins involved in signalling pathways, including FBOX3, a E3 ubiquitin-ligase, involved in SG disassembly dynamics.

## Results

### Compartmentalisation of translation-competent cytosolic extracts in water-in-oil droplets

We set out to reconstitute the sequence of events leading to SG condensation in a tuneable system that maintains near-physiological conditions. A cell-free system based on cytosolic extracts (CEs) was generated from different cell types. CE translation competence was assessed using capped, poly(A)-tailed transcripts encoding for firefly and Renilla luciferase (Fig. 1a and Supplementary Fig. 1a). CEs were treated with cycloheximide, an inhibitor of translation elongation[55], or GMP-PNP, a non-hydrolysable GTP analogue that blocks GTP hydrolysis by the eIF2 ternary complex and ribosomal subunit joining[56] as control. The translation levels of all CEs was in the range of available commercial HeLa extracts, a well-established cell-free protein synthesis system (Supplementary Fig. 1a), with only moderate variations between the different batches produced (Supplementary Fig. 1b). Reactions were carried out at the physiological temperature of 37 °C, despite slightly lower translation levels than at 30 °C, the temperature recommended for commercial cell-free systems (Fig. 1b). We further focused on CEs derived from two human cell lines: the osteosarcoma-derived U2OS cells, widely used to study SG formation[36], and the hepatocarcinoma-derived Huh7 cells, for which we previously quantitatively characterised the main ISR key components[57].

In order to recreate a small, confined environment that reflects the congested (aka crowded) viscous cytosol[58,59] and prevent water evaporation, translation-competent CEs were compartmentalised into surfactant-stabilised water-in-oil emulsion droplets−hereafter referred to as cytosolic extract-in-oil droplets (CEODs). The generation of variable-sized CEODs was achieved through the utilisation of mechanical agitation to assess possible effects of compartment size on the process[60]. The median diameter of the majority of CEODs ($25.3 \pm 13.1\,\mu m$) was comparable to that of U2OS and Huh7 cells (Supplementary Fig. 1c, d and Supplementary Table 1).

We then evaluated a number of parameters to characterise the CEODs, including coalescence, translation levels and changes of conditions over time. Minimal coalescence was observed at 37 °C for 8 h, indicating that the salt concentration of the encapsulated CEs and the reaction temperature had only a marginal effect on the emulsion stability (Supplementary Fig. 1e). The distribution of content in CEODs was evaluated by encapsulating CEs from U2OS cells stably expressing YFP-G3BP1. The distribution was uniform, as shown by a positive correlation between YFP-G3BP1 mean fluorescence intensity and CEOD diameter (Supplementary Fig. 1f). Although the method used to generate CEs induced mild hypotonic stress, as evidenced by increased phosphorylation of p38, HSP27 and eIF2α (Supplementary Fig. 1g, h),

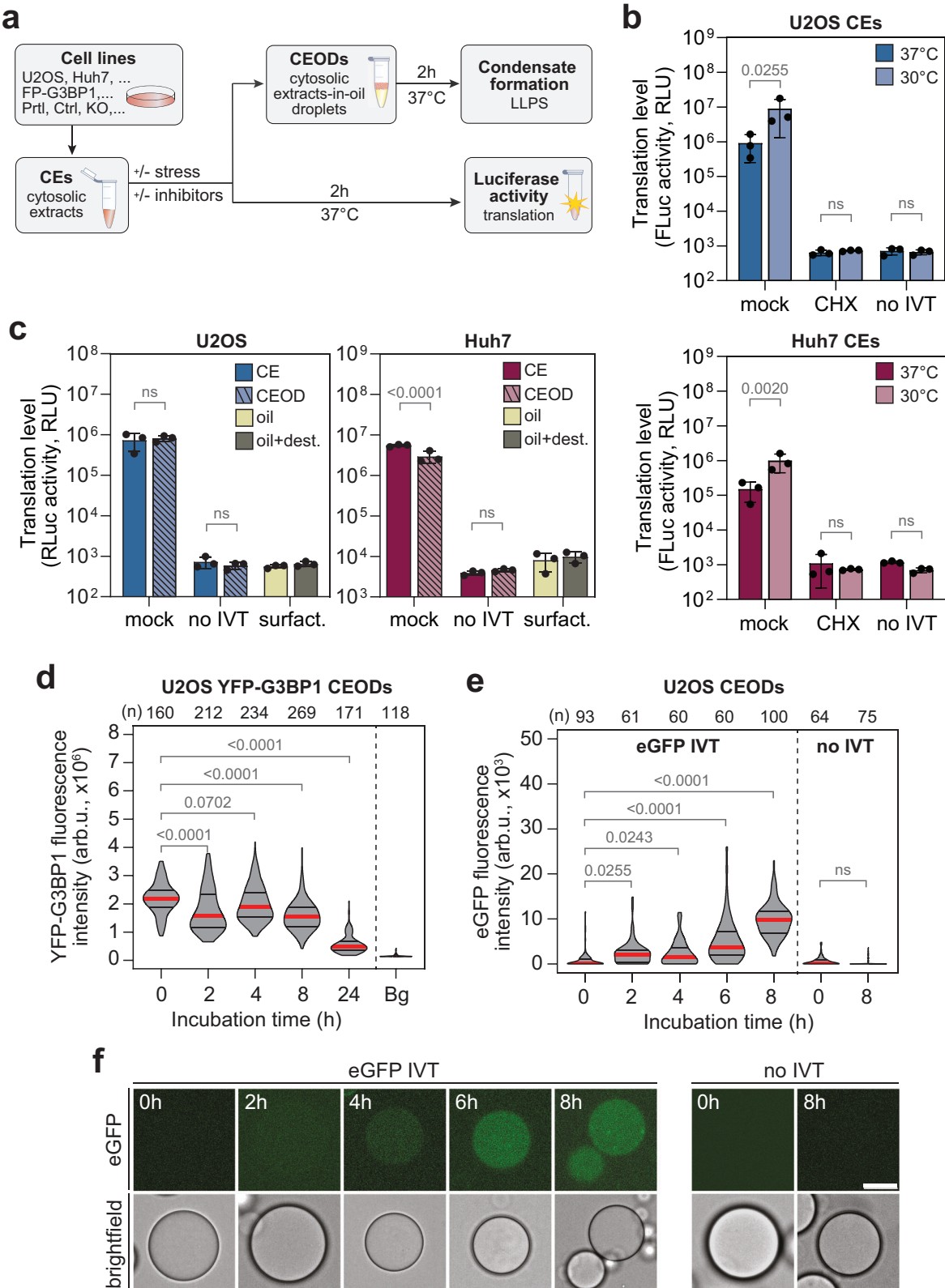

encapsulation in CEODs neither exacerbated this stress nor triggered spontaneous G3BP1 crowding, as assessed by fluorescence recovery after photobleaching (FRAP) experiments (Supplementary Fig. 1i).

We next assessed how the surfactant-stabilised oil, used to produce CEODs, influences the translation process (Fig. 1c). Following the translation reaction, the content of the CEODs was released using a destabilising agent and subsequently analysed. The translation level in

CEODs was comparable to that in CEs, indicating that the encapsulation reagents did not adversely impact on the translation process. Finally, we evaluated the stability of the conditions within CEODs at 37 °C. Residual proteolytic activity in CEs was measured by monitoring changes in YFP-G3BP1 fluorescence intensity over several hours after transferring the CEODs from 4 to 37 °C (Fig. 1d). The YFP-G3BP1 fluorescence intensity decreased during the first two hours of

**Fig. 1 | Production of translation-competent CEODs. a** Schematic of the experimental setup. **b** Translation level of a firefly luciferase (FLuc) reporter transcript in U2OS CEs (top panel) and Huh7 CEs (bottom panel), measured at 30 and 37 °C (*n* = 3 independent biological replicates). Values represent luciferase bioluminescence activity (mean ± SD), expressed as relative light units (RLU). CEs incubated without reporter transcript (no IVT) or treated with cycloheximide (CHX) served as negative control. Statistical significance (2-way ANOVA, Sidak's multiple comparisons test) compared to 37 °C is indicated. ns non-significant ($p \geq 0.05$). **c** Impact of encapsulation on translation levels (*n* = 3 independent biological replicates). Translation level (mean ± SD) of a *Renilla* luciferase (RLuc) reporter transcript in U2OS and Huh7 CEs and CEODs. The autoluminescence of surfactants reagents (oil and stabilising agents) was measured as background control. Statistical significance (2-way ANOVA, Sidak's multiple comparisons test) compared to CE is indicated. ns non-significant ($p \geq 0.05$). **d** Protein stability in CEODs. U2OS YFP-G3BP1 CEODs

were incubated at 37 °C for the indicated time period. Shown is YFP-G3BP1 fluorescence intensity (arb.u., arbitrary units) with the median indicated by a red line and the quartiles by black lines. The number of analysed CEODs (*n*) from 2 independent biological repeats and statistical significance (Kruskal–Wallis test, Dunn's multiple comparison test) compared to time 0 h are indicated. **e**, **f** Translation of an eGFP reporter transcript (eGFP IVT) in U2OS CEODs over 8 h at 37 °C. **e** Quantification of eGFP signal intensity in CEODs (arb.u., arbitrary units) with the median indicated by a red line and the quartiles by black lines. CEODs generated without eGFP reporter RNA (no IVT) were used as background control. The number of analysed CEODs (*n*) from 2 independent biological repeats and statistical significance (Kruskal–Wallis test, Dunn's multiple comparison test) compared to time 0 h are indicated. ns non-significant ($p \geq 0.05$). **f** Representative images (scale bar, 20 μm).

---

incubation at 37 °C, but remained relatively stable thereafter for up to 8 h. Real-time monitoring of eGFP synthesis, which is sensitive to large fluctuations in pH and oxygen[61,62], showed a gradual increase in fluorescence intensity over an 8-h period (Fig. 1e, f). These results suggest that CEODs undergo minimal environmental changes and proteolytic degradation. Altogether, we established a compartmentalised cell-free protein synthesis system that preserves stable, near-physiological conditions for up to 8 h.

### ISR-mediated translational repression induces G3BP1 phase separation in CEODs

To assess the stress responsiveness and SG-associated LLPS in CEODs, U2OS and Huh7 CEs were treated with double-stranded (ds) RNA, in a concentration range that we previously determined as optimal for PKR activation[57]. Single-stranded (ss) RNA and dsDNA of the same sequence and length served as negative control. In contrast to treatment with ssRNA and dsDNA, dsRNA repressed protein synthesis, with the strongest effect at 250 nM (Fig. 2a). Compared to U2OS CEs, Huh7 CEs exhibited a greater sensitivity to dsRNA, likely due to lower basal PKR levels (Supplementary Fig. 2a). Translation repression at 250 nM dsRNA was associated with a significant increase in PKR and eIF2α phosphorylation in CEs (Fig. 2b, c) and was comparable in CE and CEODs (Fig. 2d). To assess the direct involvement of PKR in translational repression, we generated CEs from U2OS ΔPKR cells[36] (Supplementary Fig. 2b) and Huh7 PKR knockout (PKR$_{KO}$) single-cell clones (Supplementary Fig. 2c, d). Treatment with dsRNA did not repress translation in CEs lacking PKR (Fig. 2e and Supplementary Fig. 2e), demonstrating a direct involvement of PKR in dsRNA-mediated ISR activation and translational repression.

We next evaluated whether the translational repression induced by treatment with dsRNAs promotes G3BP1 condensation in CEODs. To visualise this event, we generated CEODs from U2OS and Huh7 cells stably expressing YFP-G3BP1. Before encapsulation, CEs were spiked with recombinant mScarlet, a red fluorescent protein predicted to have low abundance of IDRs (Supplementary Fig. 3a)[63] and treated with dsRNA, ssRNA, or left untreated (mock) for up to 2 h at 37 °C. PEG was used as stress-independent LLPS trigger. Both dsRNA treatment and PEG induced the formation of YFP-G3BP1 condensates (Fig. 3a). To quantitatively characterise them, we devised a 2-D image analysis pipeline (Supplementary Fig. 3b). We compared: (i) the percentage of CEODs containing condensates (Fig. 3b), (ii) the condensate area for each CEODs (as measure of their abundance) (Fig. 3c, d), and (iii) the condensate perimeter and circularity (as measure of size and shape, respectively) (Fig. 3e–g). The latter was reported as the median of two biological replicates to accurately reflect variability within the heterogeneous CEOD population, with similar results observed using the mean of individual replicates (Fig. 3d).

YFP-G3BP1 condensates did not form in CEODs after a 2-h incubation at 37 °C (mock), they exhibited numerous bright, spherical YFP-G3BP1 foci (perimeter ≤ 15 μm) (Fig. 3a–c and Supplementary

Fig. 4a–e), consistent with the existence of G3BP1-containing mRNPs ("SG seeds") under normal conditions[25,35,54]. YFP-G3BP1 condensates formed upon dsRNA treatment at concentrations that repressed translation levels by at least 50% (Fig. 2a)−250 mM for U2OS CEs and 100 nM for Huh7 CEs (Fig. 3b–d and Supplementary Fig. 4). While differences in PKR expression levels may partly account for this difference, modestly higher levels of SG markers such as Caprin1, USP10 and Upf1 in Huh7 cells are also likely to contribute to the reduced threshold for LLPS (Supplementary Fig. 2a). On average, dsRNA-induced condensates formed only after approximately one hour of incubation at 37 °C, consistent with the temporal dynamics of ISR activation (Supplementary Movie 1). In contrast, PEG-induced condensates appeared immediately upon the addition of 1% PEG, even at 4 °C (Supplementary Fig. 5), underscoring that PEG and dsRNA employ distinct mechanisms to drive LLPS. Importantly, condensate formation required a threshold protein concentration of 5 μg/μl (Supplementary Fig. 6). This concentration was used in subsequent experiments, where the condensates, although less circular than SG induced in cells transfected with dsRNA, were comparable in size, with perimeters ranging from 15 to 34 μm (Fig. 3e–g and Supplementary Table 2).

Finally, we confirmed the direct involvement of PKR in translation repression by depleting PKR (Fig. 4a–d) and co-treating CEs and CEODs with ISRIB, a small molecule inhibitor of the ISR[64,65] (Fig. 4e–h). Both approaches reversed dsRNA-induced translational repression and YFP-G3BP1 condensation.

Taken together, we established a confined cell-free system that maintains stress responsiveness under near-physiological conditions, where G3BP1 LLPS occurs in sequential steps driven by ISR activation and subsequent translational repression.

### G3BP1 condensation in response to treatment with eIF4A inhibitors in CEODs

SGs also form independently of ISR activation when the eIF4F cap binding complex is altered by inhibitors targeting the eIF4A helicase activity. Compounds such as hippuristanol, pateamine A and rocaglates prevent mRNA scanning by the 43S pre-initiation complex required for translation initiation[66], thereby promoting intermolecular RNA-RNA interactions and SG formation[5,6,67]. To test whether such ISR-independent G3BP1 LLPS could also be recapitulated in CEODs, we used the rocaglate-derivative silvestrol[68], previously reported to repress translation in cell-free systems[69] and to induce SG formation in cells[70]. YFP-G3BP1 condensates formed upon treatment with 2 μg/ml silvestrol, a concentration resulting in approximately 50% translational repression in CEs (Fig. 5a–c and Supplementary Fig. 7a, b) and slightly higher than that required to induce SG formation in cells (Supplementary Fig. 7c). Shape and size of condensates were comparable to those induced by dsRNA treatment and to silvestrol-induced SGs in cells (Fig. 5d, Supplementary Fig. 7d, and Supplementary Table 2). CEODs are therefore also suitable for studying ISR-dependent and independent SG phase separation.

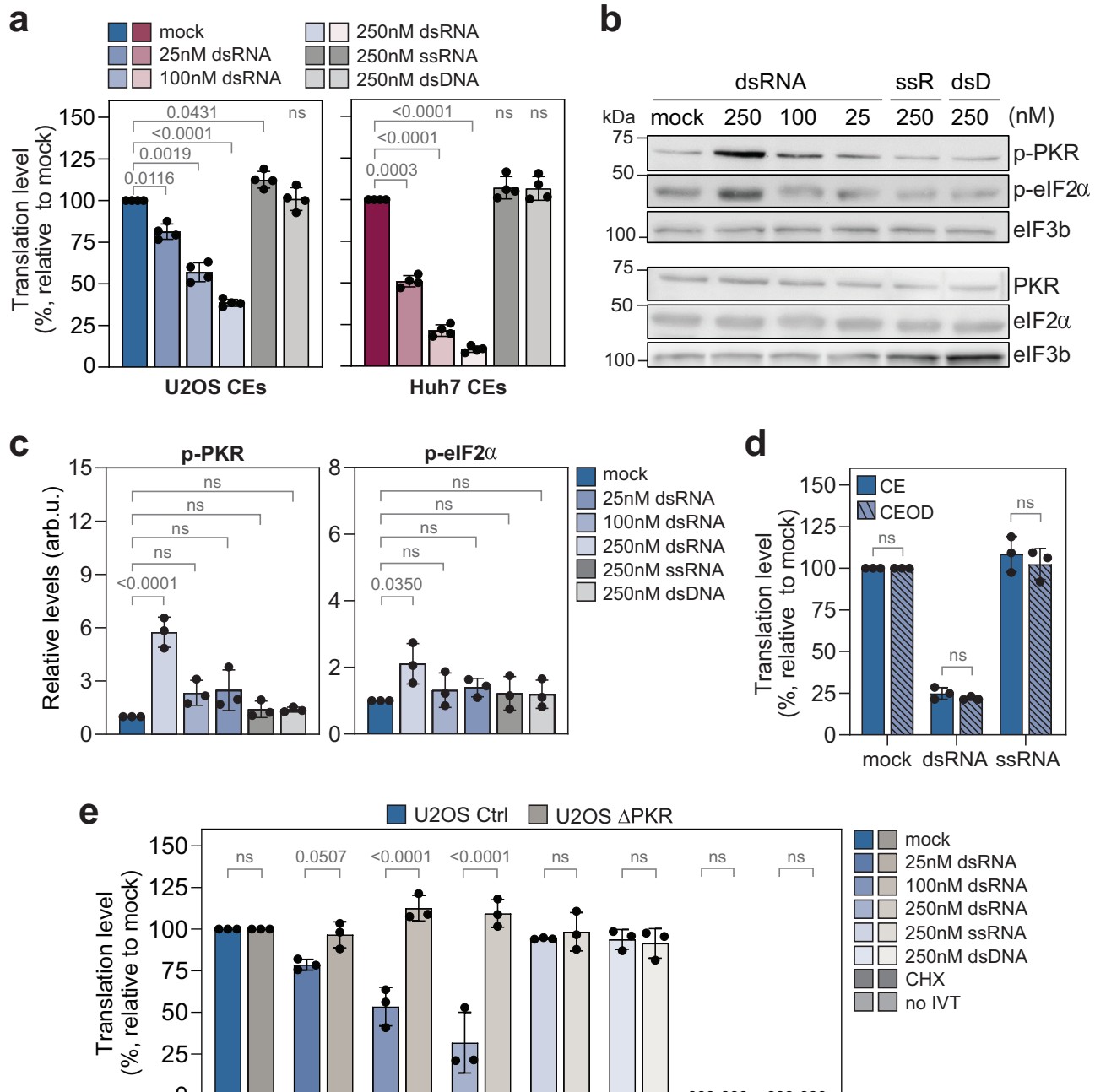

**Fig. 2 | PKR-mediated translation repression in CEs. a** Relative translation level in U2OS and Huh7 CEs treated with increasing concentrations of dsRNA ($n = 4$ independent biological replicates). Values represent percentages (mean ± SD) relative to untreated (mock) CEs. CEs treated with ssRNA and dsDNA were used as control. Statistical significance (1-way ANOVA, Dunnett's multiple comparisons test) compared to mock is indicated. **b, c** ISR activation in U2OS CEs ($n = 3$ independent biological replicates). Analysis of PKR and eIF2α basal and phosphorylated (p-) form expression levels (mean ± SD) in CEs treated with the indicated concentrations of dsRNA, ssRNA (ssR) and dsDNA (dsD). **b** Representative Western blot. Uncropped scans are provided in the Supplementary Information file. **c** Quantification of p-PKR and p-eIF2α protein levels (arb.u., arbitrary units). Basal and phosphorylated proteins were loaded on separate gels. Protein levels were normalised to their respective loading control eIF3b. Statistical significance (1-way ANOVA, Dunnett's multiple comparisons test) compared to mock is indicated. ns non-significant ($p \geq 0.05$). **d** Comparison of dsRNA-induced translation repression (mean ± SD) in U2OS CE and CEODs ($n = 3$ independent biological replicates). Statistical significance (2-way ANOVA, Sidak's multiple comparisons test) compared to CE is indicated. **e** Relative translation level of RLuc transcript in U2OS Ctrl and U2OS ΔPKR CEs treated with increasing concentrations of dsRNA ($n = 3$ independent biological replicates). Values represent percentages (mean ± SD) relative to mock CEs. Statistical significance (2-way ANOVA, Sidak's multiple comparisons test) compared to U2OS Ctrl is indicated. ns non-significant ($p \geq 0.05$).

## Condensation in CEODs requires G3BPs and cellular RNAs

Homotypic and heterotypic interactions between G3BPs and mRNAs are key driving forces of SG formation in cells[20]. We examined their contribution in the formation of stress-induced condensates in CEODs. First, we used U2OS G3BPs double KO cells (ΔΔG1/2)[19] and U2OS ΔΔG1/2 cells reconstituted with mCherry-G3BP1 (ΔΔG1/2-mChG1) as positive control. To facilitate the visualisation of phase separation in the absence of G3BPs, YFP-TIA1 was stably expressed in all cell lines. As expected, all cell lines responded to dsRNA with PKR and eIF2α phosphorylation

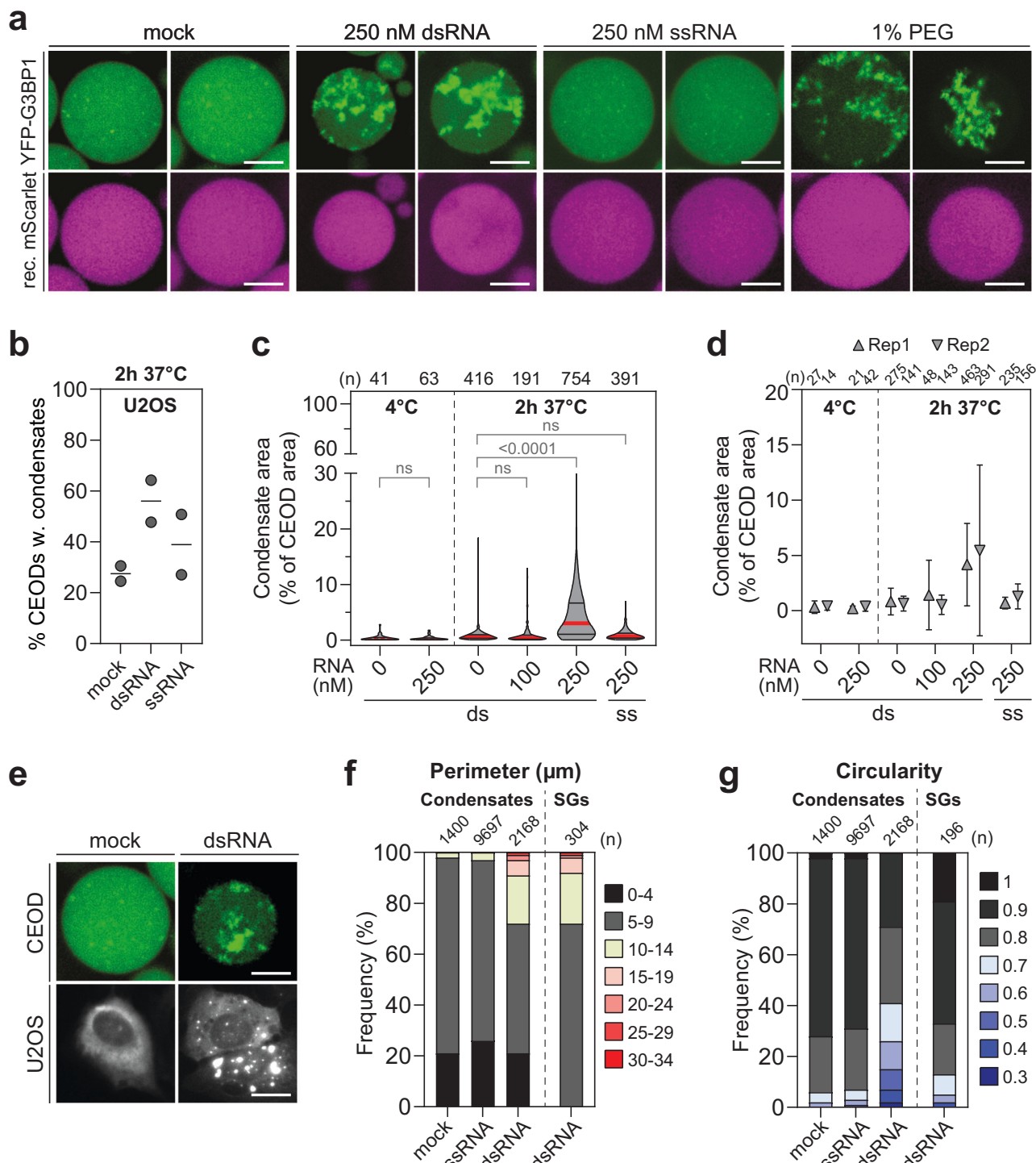

**Fig. 3 | dsRNA induces YFP-G3BP1 condensation in U2OS CEODs.**
**a** Representative images of U2OS YFP-G3BP1 CEODs. CEs were spiked with recombinant (rec.) mScarlet and treated with 250 nM dsRNA, 250 nM ssRNA or 1% PEG prior to incubation at 37 °C. Scale bar, 20 µm. **b–d** Quantification of YFP-G3BP1 condensates in response to dsRNA. CEODs treated with ssRNA were used as control. **b** Percentage of CEODs with condensates (2 biological replicates are shown). **c** Analysis of YFP-G3BP1 condensates formed at 4 and 37 °C. Shown are the median condensate area, i.e., percentage of the CEOD area covered by condensates, indicated by a red line and the quartiles indicated by black lines. The number of analysed CEODs (*n*) from 2 biological replicates is indicated on the top. Statistical significance (Kruskal–Wallis test, Dunn's multiple comparison test) compared to 0 nM dsRNA is indicated. ns non-significant (p ≥ 0.05). **d** Condensate area (mean ± SD) for each biological replicate (Rep). The number of analysed CEODs (*n*) is indicated on the top. **e–g** Comparison of dsRNA-induced condensates and SGs. **e** Representative images of dsRNA-induced YFP-G3BP1 condensates in U2OS CEODs (top panel) and dsRNA-induced SGs in YFP-G3BP1 U2OS cells (bottom panel). Scale bar, 20 µm. Bar graphs show the frequency of condensates and SGs with specified perimeter (**f**) and circularity (**g**). The number of analysed CEODs and SGs (*n*) from 2 biological replicates is indicated on the top.

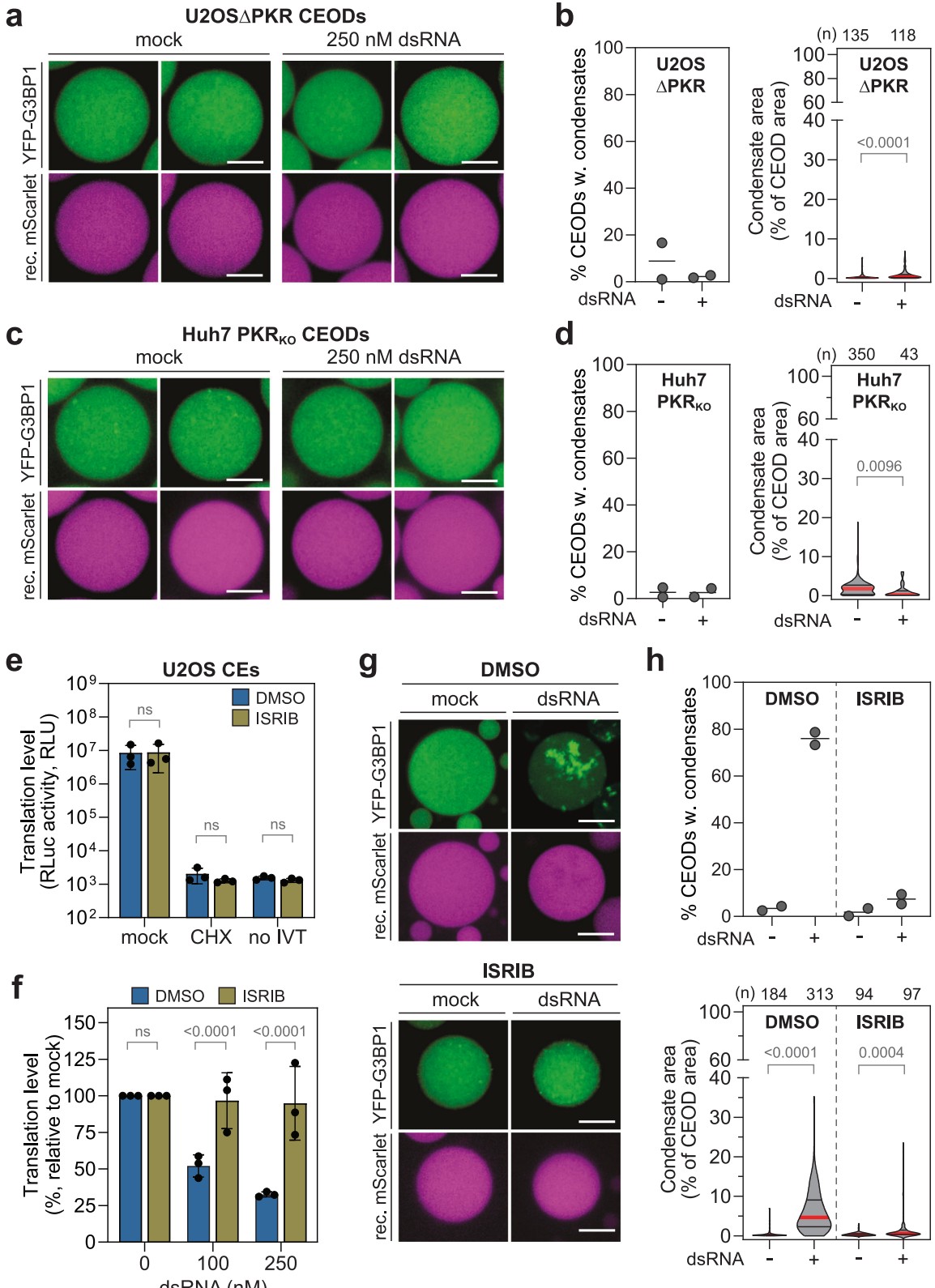

(Supplementary Fig. 8a) and depletion of G3BPs prevented SG formation, which was restored by reintroduction of mCherry-G3BP1 (Supplementary Fig. 8b). All CEs showed comparable translation levels (Supplementary Fig. 8c) and translational repression upon either dsRNA or silvestrol treatment (Supplementary Fig. 8d). Both treatments induced the formation of YFP-TIA1 condensates in U2OS WT and ΔΔG1/2-mChG1 CEODs but not in ΔΔG1/2 CEODs (Fig. 6a, b and

Supplementary Fig. 8e), confirming that G3BPs act as essential nucleators in CEODs.

Next, we assessed the importance of endogenous RNAs as scaffold molecules promoting LLPS by treating CEs with micrococcal nuclease prior to CEOD generation. Translational repression induced by dsRNA and silvestrol was unaffected by the nuclease treatment (Fig. 6c, d). However, YFP-G3BP1 condensate formation was inhibited,

**Fig. 4 | Activation of the ISR triggers YFP-G3BP1 condensate formation in CEODs.** U2OS YFP-G3BP1 ΔPKR and Huh7 YFP-G3BP1 PKR$_{KO}$ CEODs were treated with dsRNA or left untreated (mock). **a** Representative images of U2OS ΔPKR CEODs. Scale bar, 20 μm. **b** Left panel: percentage of U2OS ΔPKR CEODs with condensates (2 biological replicates are shown). Right panel: corresponding YFP-G3BP1 condensate area with the median indicated by a red line and the quartiles by black lines. The number of analysed CEODs (*n*) from 2 biological replicates and statistical significance (Kruskal−Wallis test, Dunn's multiple comparison test) compared to−dsRNA are indicated. **c, d** Huh7 PKR$_{KO}$ CEODs and corresponding condensate analysis. **c** Representative images of Huh7 PKR$_{KO}$ CEODs. Scale bar, 20 μm. **d** Left panel: percentage of Huh7 PKR$_{KO}$ CEODs with condensates (2 biological replicates are shown). Right panel: shown is the corresponding YFP-G3BP1 condensate area with the median indicated by a red line and the quartiles by black lines. The number of analysed CEODs (*n*) from 2 biological replicates and statistical significance (Kruskal−Wallis test, Dunn's multiple comparison test) compared to− dsRNA are indicated. **e–h** Impact of ISRIB on the translation in U2OS CEs (*n* = 3 independent biological replicates). (**e**) Translation level (mean ± SD) of RLuc transcript in U2OS CEs treated with ISRIB or vehicle control DMSO. Statistical significance (2-way ANOVA, Sidak's multiple comparisons test) compared to DMSO is indicated. **f** Relative translation levels upon treatment with dsRNA. Values (mean ± SD) are represented as percentage relative to untreated CEs (0 nM dsRNA). Statistical significance (2-way ANOVA, Sidak's multiple comparisons test) compared to DMSO is indicated. ns non-significant (*p* ≥ 0.05). **g** Representative images of CEODs treated with dsRNA and ISRIB or DMSO. Scale bar, 20 μm. **h** Top panel: percentage of CEODs with condensates (2 biological replicates are shown). Bottom panel: shown is the corresponding YFP-G3BP1 condensate area with the median indicated by a red line and the quartiles by black lines. The number of analysed CEODs (*n*) from 2 biological replicates and statistical significance (Kruskal−Wallis test, Dunn's multiple comparison test) compared to−dsRNA are indicated. ns non-significant (*p* ≥ 0.05).

including in PEG-treated CEODs (Fig. 6e, f and Supplementary Fig. 9a, b). Overall, these results highlighted that, as in cells, G3BPs and mRNAs are essential scaffold molecules driving phase separation in response to translation repression in CEODs.

## The USP10 peptide inhibits condensate formation in CEODs

The ubiquitin-specific peptidase 10 (USP10) represses SG formation by binding G3BP1 through its Phe-Gly-Asp-Phe motif (FGDF, aa 10–13), thereby displacing Caprin1[19,71]. USP10 limits G3BP1 valency with other RBPs[22]. To assess if such regulation is also reflected in our system, we asked whether an USP10-derived peptide containing the FGDF motif (USP10$_{8-25}$)[19] prevents YFP-G3BP1 condensation. To this end, we estimated the intracellular concentration of YFP-G3BP1 in U2OS YFP-G3BP1 cells using quantitative Western blotting (Supplementary Fig. 10a, b). U2OS CEs were incubated with a 10- and 100-fold excess (1 and 10 μM, respectively) of wild type (WT) or a mutant peptide in which both phenylalanine residues were substituted by alanine residues (F10A−F13A). Consistent with previous findings in cells showing that USP10-mediated repression of SG formation occurs downstream of polysome disassembly[19], the addition of the USP10-derived peptides did not affect CE translation levels (Supplementary Fig. 10c) or the levels of translation repression in response to dsRNA (Supplementary Fig. 10d). However, the WT peptide, but not the F10A-F13A mutant peptide, inhibited YFP-G3BP1 condensation in CEODs (Fig. 7a, b). Collectively, these results demonstrated that YFP-G3BP1 phase separation can be inhibited in CEODs.

## Mobility of G3BP1 in condensates

SGs comprise numerous dynamic components, including G3BP1, which are in constant exchange with the cytosol[72]. We measured the mobility of mCherry-G3BP1 within the different types of condensates formed in CEODs using FRAP (Fig. 8a). In dsRNA-induced condensates, the recovery of the mCherry-G3BP1 signal had a half-life time of 4.8 s ± 0.6 (Fig. 8b and Supplementary Movie 2), a dynamics in line with previous findings in cells[21,31] and in G3BP1-induced granules in lysates[52]. Comparable mobility was measured in PEG-induced condensates ($t_{1/2}$ = 5.7 s ± 0.7). Remarkably, the recovery of the mCherry-G3BP1 signal in silvestrol-induced condensates was markedly slower ($t_{1/2}$ = 99.4 s ± 28.3), corroborating previous observations that, in cells, pateamine A-induced SGs are less prone to disassembly than those formed in response to arsenite[5]. This difference was not attributable to changes in protein solubility (Supplementary Fig. 11a) but likely reflects distinct biophysical properties of silvestrol and dsRNA-induced condensates, possibly due to silvestrol's ability to clamp onto larger RNA regions[73].

## G3BP1 phase separation triggers ultrastructural changes in CEODs

To assess condensate organisation and density at the ultrasutructural level, CEODs were freeze-fractured following the reaction and analysed by cryo-scanning electron microscopy (cryo-SEM)[74] (Supplementary Fig. 11b). Of note, control mock and ssRNA CEODs exhibited a striking difference in stability compared to the CEODs undergoing phase separation and did not withstand nitrogen plunge freezing after incubation at 37 °C. This observation indicated possible changes in the surface tension between control and phase-separated CEODs and confirmed a remodelling of the CEOD content upontranslation inhibition. As increasing the surfactant concentration did not overcome this issue, we lowered the reaction temperature for the control CEODs to room temperature. Compared to the control CEODs whose content showed a fine mesh structure (Supplementary Fig. 11b), CEODs treated with dsRNA or silvestrol displayed thin sheets of condensed proteins, resembling the microgels formed from nanofibril-forming proteins within water-in-oil droplets[75]. CEODs treated with PEG showed similar but thicker structures. Although qualitative, these results demonstrated that the inhibition of translation induced by dsRNA or silvestrol leads to a significant partitioning of the entire cytosolic content of the CEODs into a network of condensed proteins.

## The RNA composition of SG-like condensates is similar to that of cellular SGs

To evaluate the RNA composition, YFP-G3BP1 condensates induced by dsRNA, silvestrol, PEG treatments or untreated (mock) were isolated by pelleting, total RNA was extracted, and poly(A) mRNAs subsequently analyzed by RNA sequencing. Poly(A) mRNAs extracted from an equivalent amount of input CE served as a reference (Supplementary Data 1). RNAs enriched in condensates included about one-third of the RNAs previously identified in the transcriptome of arsenite-induced SGs[38] (Fig. 8c and Supplementary Fig. 12a). Condensate transcriptomes showed a similar RNA enrichment-depletion relative to total RNA than observed in arsenite-induced SGs (Supplementary Data 2 and Fig. 8d), with longer transcripts (e.g., *AHNAK2*, *BCL9L*, *MAP1A*, *AHNAK*, *DYNC1H1*, *PEG3*, *ZNF704*, *CDK6*) being preferentially enriched compared with shorter ones (e.g., *RLP39*, *RPS29*, *RPL31*, *GAPDH*) (Fig. 8e, f). A comparable RNA distribution was observed in mock condensates, suggesting that a subset of RNAs enriched in condensates is likely pre-associated with mRNPs prior to translation repression and LLPS (Fig. 8c–f). Reinforcing this notion, comparative analysis between mock, dsRNA- or silvesterol-induced condensate transcriptomes revealed no significant differences in their RNA compostion (Supplementary Fig. 12b). The RNA composition of PEG-induced condensates differed markedly from other conditions: fewer than 1% overlapped with the transcriptome of arsenite-induced SGs (Fig. 8c), and shorter RNAs were enriched over longer ones (Fig. 8e, f), underscoring the fundamental distinction between LLPS driven by molecular crowders and that triggered by translational repression. Overall, these analyses demonstrate that the RNA content of the condensates induced by translation repression in the CEOD system recapitulates in part the RNA composition of SGs assembled in cells upon arsenite treatment.

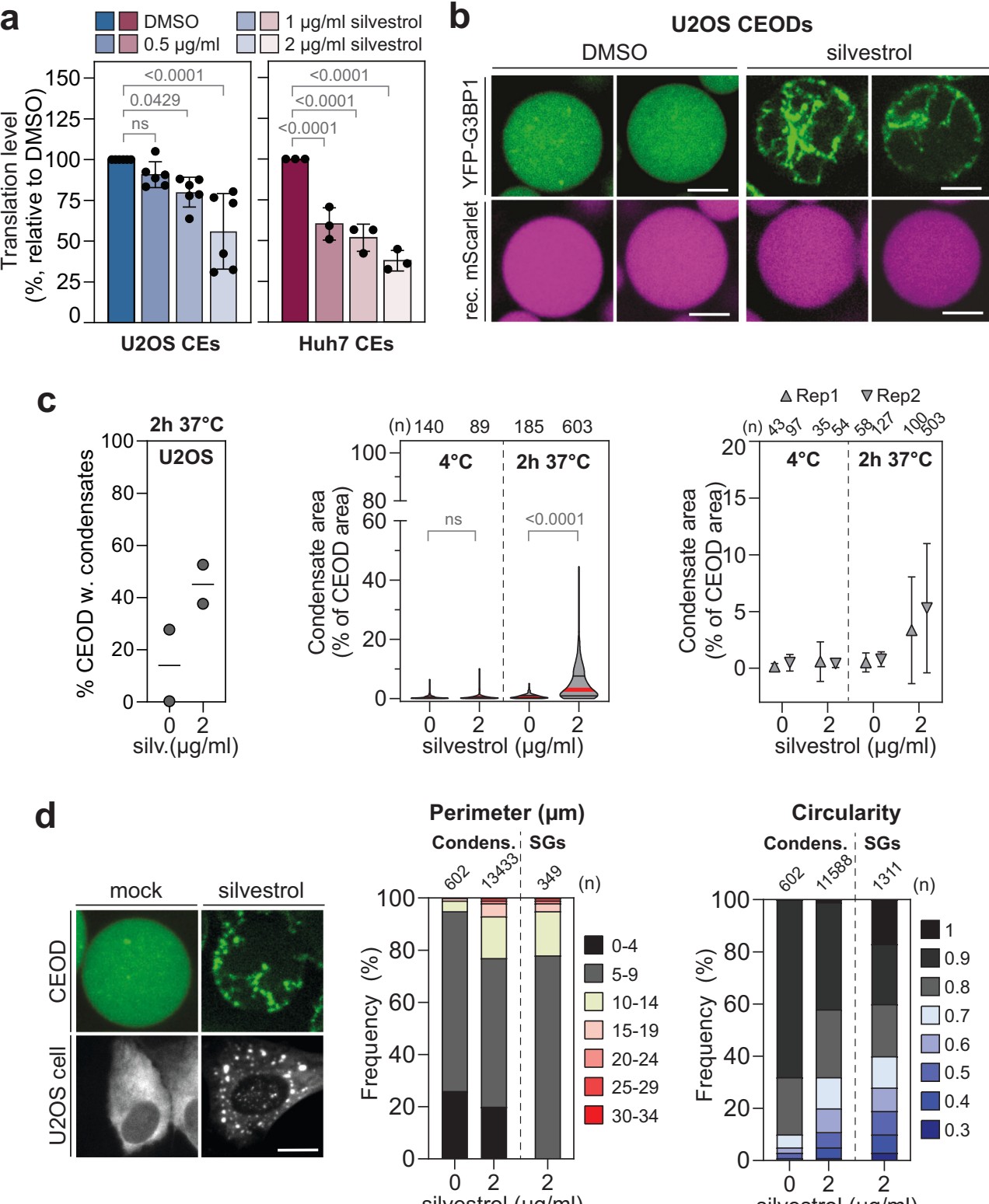

**SG-like condensate proteome identifies new SG client proteins**

The application of various techniques, including candidate-based approaches, knockdown and KO screens, and purification/labelling methods coupled to mass spectrometry, has uncovered a marked heterogeneity in the composition of SGs. The overlap was limited and varied depending on the treatment conditions and cell line[76]. Given that more dynamic components of the SG shell are often lost during enrichment or purification procedures, we postulated that our approach would facilitate the discovery of yet unidentified SG client proteins that are recruited upon ISR activation. To resolve potential compositional differences, mock, dsRNA, silvestrol or PEG condensates were subjected to an unbiased proteomic analysis using high-sensitivity label-free liquid chromatography coupled to mass spectrometry (LC-MS). Comparative analysis of normalised protein intensities of our input sample (CE proteome) with those obtained from a publicly available deep global proteome of U2OS cells (Supplementary

**Fig. 5 | YFP-G3BP1 condensates form in response to eIF4A inhibition in CEODs.**
**a** Relative translation levels of RLuc transcript in U2OS CEs ($n = 6$ biological repeats) and Huh7 CEa ($n = 3$ biological repeats) treated with increasing concentrations of silvestrol. Values (mean ± SD) are represented as percentage relative to vehicle control (DMSO). Statistical significance (1-way ANOVA, Dunnett's multiple comparisons test) compared to DMSO is indicated. ns non-significant ($p \geq 0.05$).
**b, c** Induction of YFP-G3BP1 condensation by silvestrol in CEODs. **b** Representative images of U2OS YFP-G3BP1 CEODs treated with 2 μg/ml silvestrol or DMSO. Sscale bar, 20 μm. **c** Left panel: percentage of CEODs with condensates (2 biological replicates are shown). Middle panel: shown is the corresponding YFP-G3BP1 condensate area with the median indicated by a red line and the quartiles by black lines.

The number of analysed CEODs ($n$) from 2 biological replicates and statistical significance (Kruskal–Wallis test, Dunn's multiple comparison test) compared to 0 μg/ml silvestrol are indicated. ns non-significant ($p \geq 0.05$). Right panel: YFP-G3BP1 condensate area (mean ± SD) for each biological replicate (Rep). The number of analysed CEODs for each biological replicate ($n$) is indicated on the top.
**d** Comparison of condensates in CEODs and SGs in U2OS cells treated with 2 μg/ml silvestrol. Shown are representative images (left panel). Scale bar, 20 μm. Bar graphs show the frequency of silvestrol-induced condensates and SGs with specified perimeter (middle panel) and circularity (right panel). The number of analysed CEODs and SGs ($n$) from 2 biological replicates is indicated on the top.

Fig. 13a–c) revealed a high degree of similarity, excluding potential methodological biases arising from the CE preparation method. The expression levels of the main SG proteins G3BP1 and Caprin1 in CE were comparable to those reported in the published U2OS cell proteome. USP10 expression was slightly reduced, highlighting cell line-specific differences (Supplementary Fig. 13a–c).

The quantitative proteomic analysis of pelleted condensates detected a remarkably high number of proteins. A lower but substantial amount of proteins was detected in mock samples (Supplementary Fig. 14a and Supplementary Data 3). Condensate proteomes exhibited a strong overlap accross all conditions (Supplementary Fig. 14a). Most high-confidence SG components—as defined in the curated RNA Granule Database (SG tier 1 and SG and P-bodies tier 1)[77] or identified through other methods such as lysate granules[52]—as well as 45 SG "core" components shortlisted based on their enrichment by proximity-dependent biotinylation coupled with mass spectrometry[25,35], were enriched within the condensates (Supplementary Fig. 14b). Strikingly, the mock condensate proteome included approximately 67% of SG tier 1 components and 40 out of 45 SG core components (Supplementary Fig. 14b). This finding supported the notion that pre-existing mRNPs may be present under normal conditions and serve as nucleation seeds supporting the formation of an expanded network when cells are exposed to stress[25,35,54]. While these numbers were notable, statistical analysis demonstrated that the observed enrichment reflected selective recruitment rather than random or procedural effects associated with the pelleting step.

To uncover condition-specific proteins, we performed unsupervised hierarchical clustering (Fig. 9a and Supplementary Data 3). This analysis identified four distinct clusters comprising proteins specifically enriched in each treatment group (315 proteins in dsRNA, 755 in silvestrol), as well as a larger cluster of proteins shared between the dsRNA and silvestrol treatments (2049 proteins). As anticipated, substantial differences emerged between the proteomes of condensates formed through translation inhibition and those formed via PEG-induced molecular crowding. Notably, PEG treatment generally enriched proteins abundant in the CE (input), whereas treatment with dsRNA enriched proteins that were low-abundant or undetectable in the input (Supplementary Fig. 15). Analysis of gene ontology (GO) annotations significantly enriched within individual clusters identified key SG-associated terms[77]—including RNA metabolic processes, translation, RNA stability, RNA processing and RNA splicing. Most enriched GO biological processes (GOBPs) terms in dsRNA-induced condensates were related to mitochondrial integrity maintenance, translation, and respiration, and related to mRNA stabilisation and ribosome biogenesis in silvestrol-induced condensates (Supplementary Fig. 16a and Supplementary Data 4).

We performed differential abundance analysis to further compare the condensate composition. To circumvent the issue of undetectable proteins in input and mock samples, we compared dsRNA- or silvestrol-induced condensates with PEG-induced condensates that reflected non-specific enrichment (log2 FC > 2.5). This analyses identified 147 proteins exclusively enriched in dsRNA-induced condensates

and 14 in silvestrol-induced condensates (Fig. 9b and Supplementary Data 5). We confirmed the specific enrichment of proteins related to mitochondrial maintenance and translation in dsRNA-induced condensates (Fig. 9c). The localisation of four candidates—mitochondrial genome maintenance exonuclease 1 (MGME1), mitochondrial ribosome recycling factor (MRRF), NADH:ubiquinone oxidoreductase subunit S6 (NDUFS6) and mitochondrial ribosomal protein L51 (MRPL51)—was assessed by immunofluorescence assay in CEODs. Consistent with their enrichment detected by mass spectrometry, all candidates co-localised within YFP-G3BP1 condensates induced by dsRNA but showed no or only poor co-localisation within silvestrol-induced condensates (Supplementary Fig. 16b, c). In cells, MGME1 specifically localised to dsRNA-induced SGs but not to arsenite or silvestrol-induced SGs (Fig. 9d, e). This result confirmed that CEODs enable the identification of dsRNA-specific SG components and supported the notion that SG composition is heterogeneous and depends on the type of stress.

Next, we exploited the clusters of proteins enriched in condensates following both dsRNA and silvestrol treatments to identify potential SG client proteins generally enriched by translation repression. We prioritised candidates based on (i) their absence from the RNA Granule Database[39], (ii) their predicted cytosolic localisation, (iii) antibody availability, and (iv) presence of at least two proteins from the same family (Supplementary Data 5). Using this approach, we validated six host proteins as bona fide SG components. These included four E3 ubiquitin-ligases—F-Box protein 3 (FBXO3), HECT and RLD domain containing E3 ubiquitin protein ligase 3 (HERC3), tripartite motif containing 37 (TRIM37) and TRIM38—, chaperone DnaJ heat shock protein family (Hsp40) member C6 (DNAJC6), and glycosyltransferase alpha-1,6-mannosylglycoprotein 6-beta-N-acetylglucosaminyltransferase (MGAT5). Consistent with their enrichment detected by mass spectrometry, all candidates co-localised within YFP-G3BP1 condensates (Supplementary Fig. 17) and in SGs formed in response to dsRNA or silvestrol in U2OS cells (Fig. 10a, b and Supplementary Fig. 18). Interestingly, all candidates also localised into arsenite-induced SGs, indicating that their recruitment into SGs is conserved under various stress conditions.

E3 ubiquitin ligases have been linked to SG disassembly[78]. We conducted a targeted loss-of-function screen using siRNA pools to delineate the effects of candidate gene knockdown on this process (Supplementary Fig. 19a). U2OS cells were subsequently stressed either with arsenite or silvestrol and the fraction of SG-positive cells was assessed after stress removal. Silencing of the newly identified SG clients did not affect SG assembly in response to treatment with arsenite or silvestrol (0 min post drug removal, Supplementary Fig. 19a). Depletion of FBXO3 led to a modest yet significant acceleration of SG disassembly under both stress conditions (Supplementary Fig. 19a), an effect that was recapitulated in U2OS FBXO3$_{KO}$ cells compared with control cells (Ctrl) (Fig. 10c and Supplementary Fig. 19b, c). These findings suggest that FBXO3 contributes to the temporal regulation of SG dynamics by delaying their disassembly. Among the Nedd4 family HECT-type E3 ubiquitin

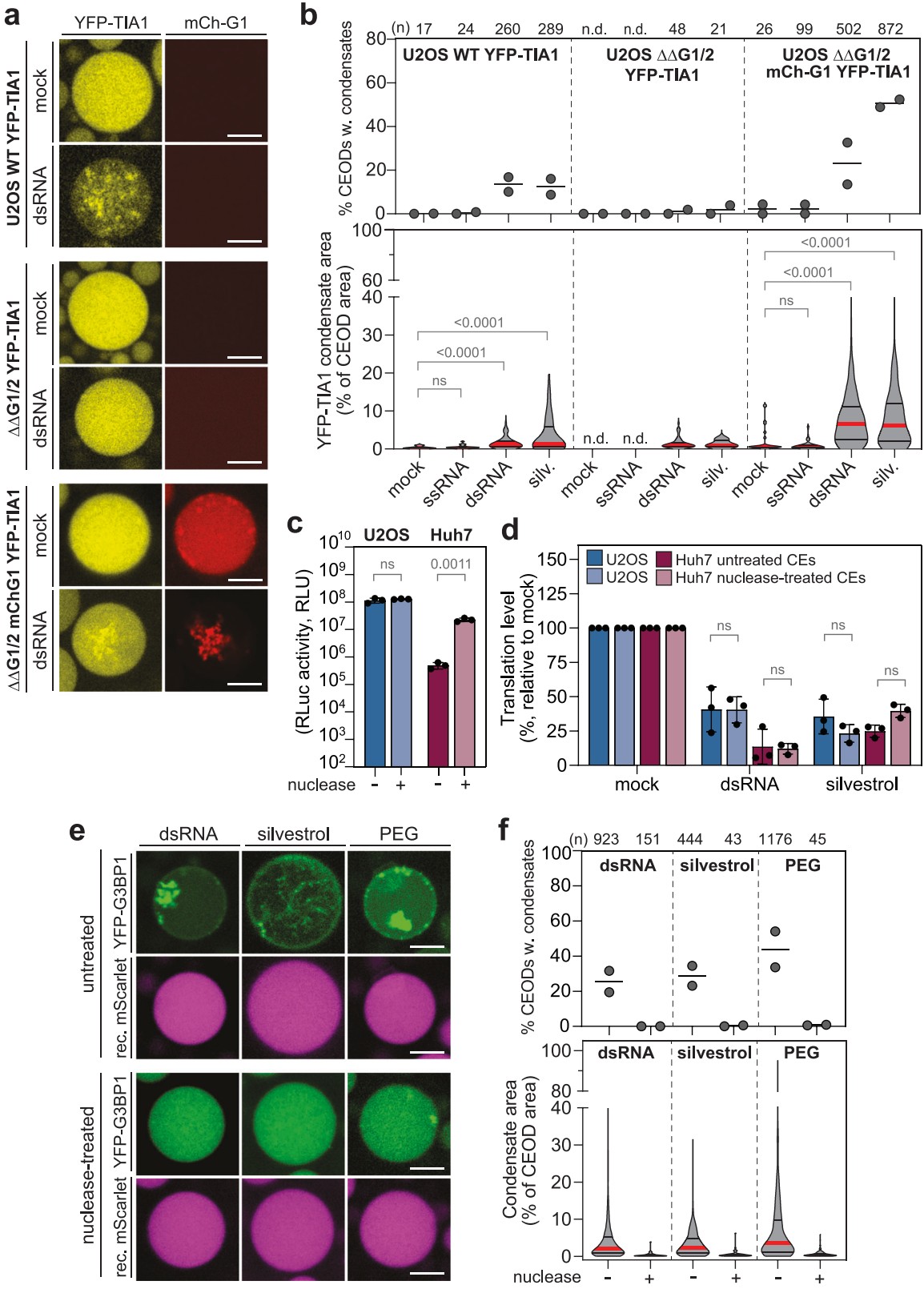

ligases reported as FBXO3 targets[79], only the small isoform of ITCH accumulated in U2OS FBXO3$_{KO}$ cells (Supplementary Fig. 19d). Consistent with this observation, ITCH was recently shown to promote SG disassembly[78], thereby exerting an effect opposite to that of FBXO3. Together, these results support a model in which a balanced interplay of E3 ligase activities ensures the precise temporal control of SG clearance.

## Discussion

In this study, we generated a unique stress-responsive cell-sized CEOD system that is translation competent and supports SG phase separation both in an eIF2α phosphorylation-dependent (i.e., in response to the ISR activation by dsRNA) and -independent manner (i.e., when interfering with the cap binding complex). This system maintains near-physiological conditions, in which cell signalling, including PTMs are

**Fig. 6 | Condensate formation in CEODs depends on G3BP1 and cellular RNAs.** The SG marker YFP-TIA1 was stably expressed in U2OS WT cells, U2OS ΔΔG1/2 cells and U2OS ΔΔG1/2 mCh-G1 cells. CEODs were generated from all cell lines and treated with dsRNA. **a** Representative images. Scale bar, 20 μm. **b** Top panel: percentage of CEODs with condensates (2 biological replicates are shown). Bottom panel: shown is the corresponding YFP-G3BP1 condensate area with the median indicated by a red line and the quartiles by black lines. The number of analysed CEODs ($n$) from 2 biological replicates and statistical significance (Kruskal–Wallis test, Dunn's multiple comparison test) compared to mock are indicated. ns non-significant ($p \geq 0.05$). **c–f** Impact of micrococcal nuclease treatment on U2OS and Huh7 YFP-G3BP1. CEs were treated with 18U micrococcal nuclease prior to the translation reaction. **c** Translation levels of RLuc transcript in U2OS and Huh7 CEs ($n = 3$ independent biological replicates). Values (mean ± SD) correspond to the

RLuc activity and are expressed as relative light units (RLU). Statistical significance (2-way ANOVA, Sidak's multiple comparisons test) compared to - nuclease is indicated. ns non-significant ($p \geq 0.05$). **d** Relative translation level upon treatment with dsRNA or silvestrol ($n = 3$ independent biological replicates). Values (mean ± SD) are represented as percentage relative to mock CEs. Statistical significance (2-way ANOVA, Tukey's multiple comparisons test) compared to untreated is indicated. ns non-significant ($p \geq 0.05$). **e** Impact of micrococcal nuclease treatment on YFP-G3BP1 condensation in U2OS CEODs treated with dsRNA, silvestrol or PEG. Shown are representative images of CEODs. Scale bar, 20 μm. **f** Top panel: corresponding quantification of the percentage of CEODs with condensates (2 biological replicates are shown). Bottom panel: shown is YFP-G3BP1 condensate area with the median indicated by a red line and the quartiles by black lines. The number of analysed CEODs ($n$) from 2 biological replicates is indicated on the top.

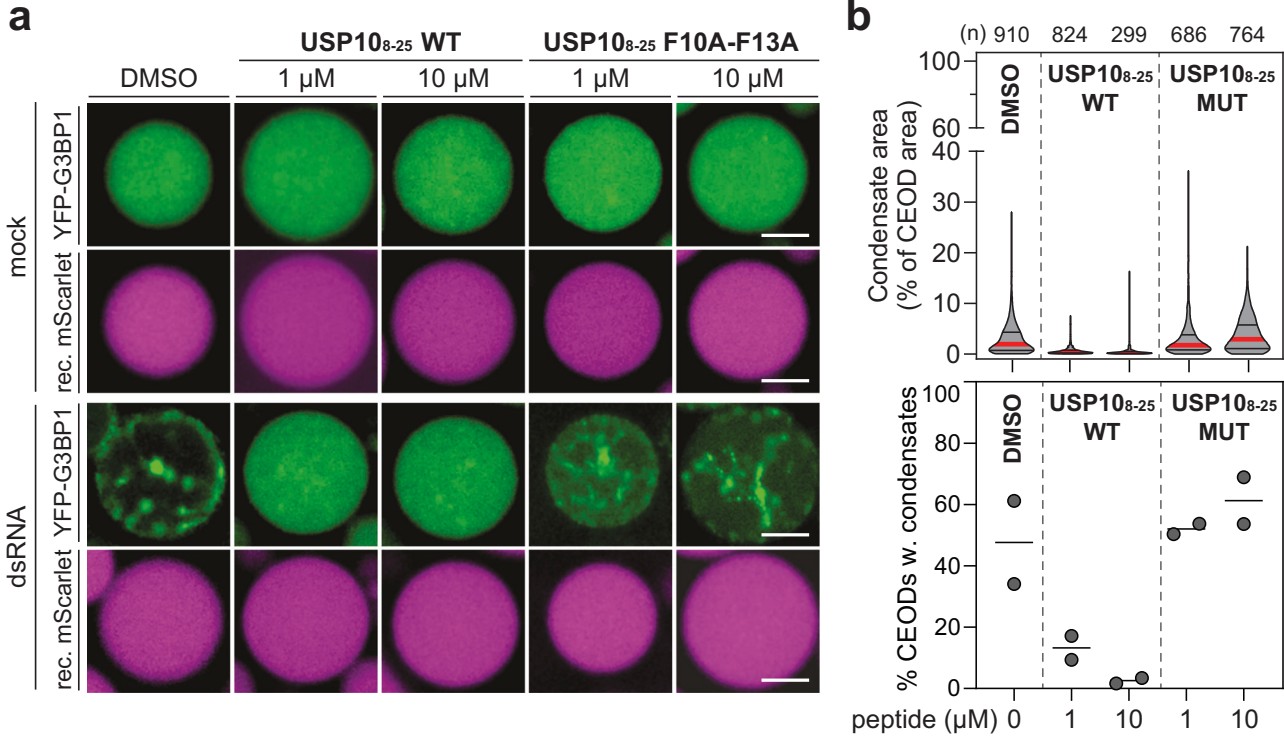

**Fig. 7 | Inhibition of G3BP1 condensation in CEODs by the USP10 peptide.** U2OS CEODs were treated with dsRNA and increasing concentration of USP10$_{8–25}$ peptide with wild-type sequence (WT) or mutated sequence (F10A–F13A). **a** Representative images of CEODs. Scale bar, 20 μm. **b** Top panel: shown is YFP-G3BP1 condensate

area with the median indicated by a red line and the quartiles by black lines. The number of analysed CEODs ($n$) from 2 biological replicates is indicated on the top. Bottom panel: percentage of CEODs with condensates (2 biological replicates are shown).

active, allowing the study of specific components in the various steps leading to SG condensation. The compartmentalisation of CEs within CEODs provides numerous advantages over conventional approaches, in terms of throughput and cost-effectiveness, sample volume, assay time, and adaptability to different cellular contexts, including human disease contexts. Furthermore, our results demonstrate the versatility and ability of this approach to infer the composition of various condensates and identify new SG client proteins, while also providing a tractable system for mechanistic and functional studies.

Previous studies have provided elegant proof-of-principles for the reconstitution and purification of SG condensates in vitro by spiking cellular extracts with high concentrations of recombinant G3BP1[52] or exogenous RNA[53] to trigger phase separation. However, these approaches did not provide insight into the intracellular signalling pathways activated during stress, including associated PTMs. For instance, PTMs such as phosphorylation, methylation, and

ubiquitination are involved in the timely assembly and disassembly of SGs and may influence their composition[80,81]. Here, we have leveraged cell-free translation systems and protocell engineering approaches[82–87] to develop an in vitro system for studying SG condensation that more faithfully resembles the process occurring in stressed cells. Conditions in CEODs remain near-physiological for at least 8 h with only minimal biodegradation, enabling protein synthesis and phenotypic analysis by confocal microscopy. Importantly, we demonstrated that CEODs are "stress responsive", i.e., capable of recapitulating the complete cascade of event leading to SG condensation in response to treatment with dsRNA: from PKR-mediated ISR activation to eIF2α phosphorylation and translation repression, followed by G3BP1-dependent phase separation. Both transcriptome and proteome analyses of the condensates showed a remarkable overlap with previously published arsenite-induced SG transcriptome[38] and RNA Granule Database tier 1, suggesting that condensates induced by translational repressors in this system can be considered as SG-like condensates.

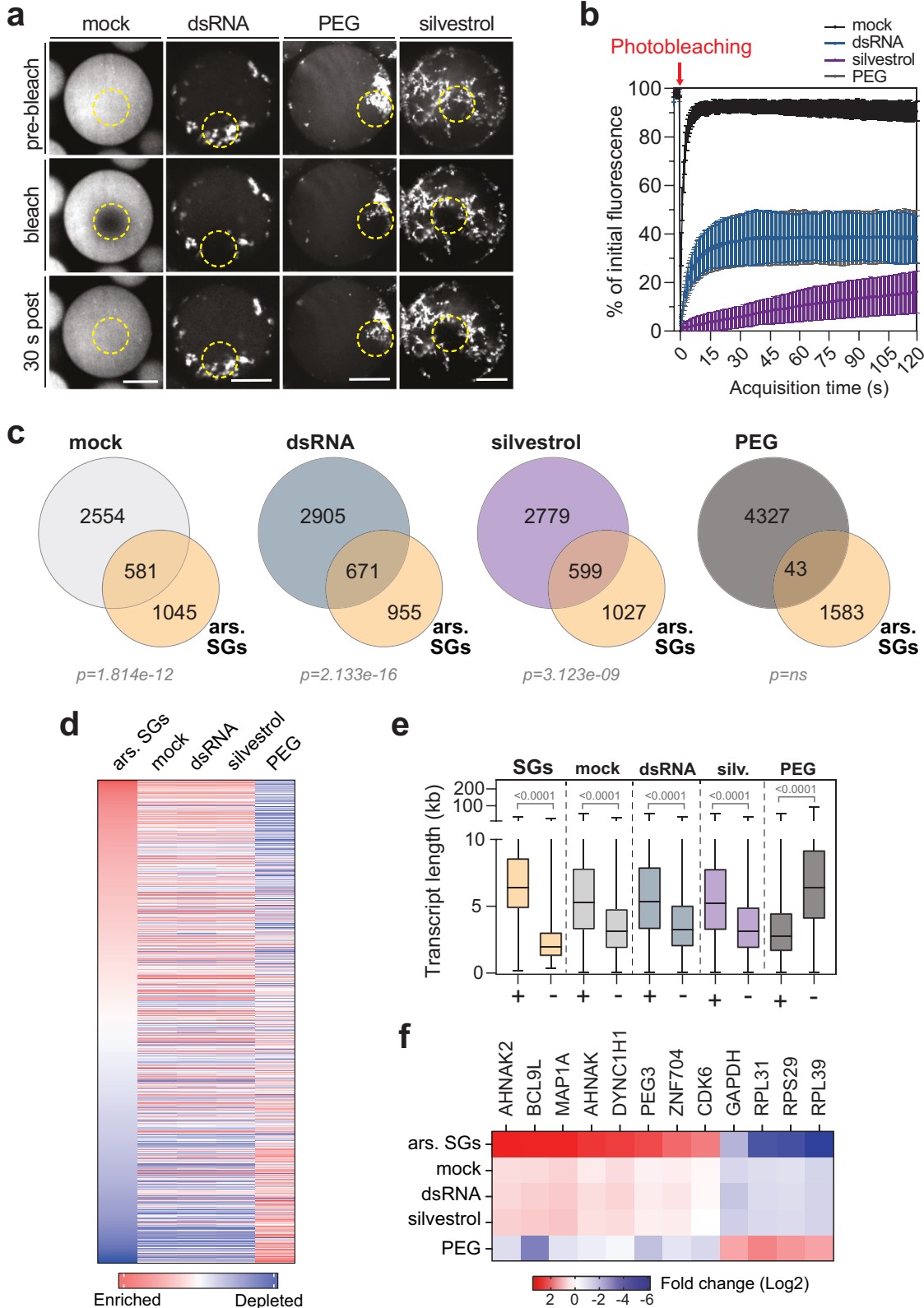

We initially attempted to trigger G3BP1 LLPS in CEODs using sodium arsenite, a well-established inducer of oxidative stress known to activate HRI and promotes SG formation[14,88]. Although arsenite repressed translation by at least 50% at high concentrations (0.5 mM for U2OS CEs and 1.5 mM for Huh7 CEs) (Supplementary Fig. 20a), YFP-G3BP1 condensates did not form (Supplementary Fig. 20b, c). By adding recombinant firefly luciferase to CEs, we found that in this

system, arsenite-induced pleiotropic toxicity directly targeted the luciferase, reducing its activity by approximately three logs (Supplementary Fig. 20d). This suggested that, as previously reported, arsenite likely affects a broad range of other cytosolic enzymes as well[89,90]. This outcome evidenced the unsuitability of arsenite as a stress inducer for the CEOD approach.

**Fig. 8 | G3BP1 dynamic in CEOD and condensate transcriptome analysis.**
**a**, **b** G3BP1 mobility in CEODs was measured by FRAP. CEODs were generated with U2OS ΔΔG1/2 mCh-G1 CEs and treated with dsRNA, silvestrol, PEG or left untreated (mock). A region of interest (yellow dotted circle) was photobleached for 5 s. The recovery of the mCherry signal was measured for 180 s. **a** Representative images before, during and after bleaching. Scale bar, 20 μm. **b** Quantification of mCherry-G3BP1 signal over time. Results are shown as percentage of the initial fluorescence before bleaching (mean ± SD). 20 condensates were analysed per condition. **c–f** Condensate transcriptomes. Condensates were induced by treatment with dsRNA, silvestrol, or PEG ($n = 3$ independent biological replicates), pelleted and mRNAs analysed by RNA-seq. **c** Venn diagrams comparing unique and overlapping mRNAs between condensates (dsRNA, silvestrol, PEG or mock) and previously published transcriptome of arsenite-induced SGs (Khong et al.[38]) (ars. SGs). mRNAs selected in each condition were identified in at least 2 replicates per condition. Statistical significance (hypergeometric enrichment test, $p$-values adjusted for multiple testing using the Benjamini–Hochberg FDR) for the overlap between the two groups is shown as adjusted $p$-values at the bottom. **d** mRNAs identified in condensates were marked as depleted or enriched over input (CE transcriptome) based on Log2 fold change. Results were rank ordered and compared to the previously published transcriptome of arsenite-induced SGs by Khong et al.[38] (ars. SGs). **e** Box plots showing the length distribution (in kilobases, kb) of mRNAs that are enriched (+, fold change >1) or depleted (−, fold change <0.6) in each type of condensates compared to arsenite-induced SGs (SGs). The median is shown by the central line, the box edges represent the 25th and 75th percentiles, and the whiskers extend to the minimum and maximum values. Statistical significance (1-way ANOVA, Tukey's multiple comparisons test) compared to enriched mRNAs (+) is indicated. **f** Differential expression analysis of selected mRNAs strongly enriched or depleted in the SG transcriptome, compared with their level in condensates. Displayed are fold changes relative to the input (CE transcriptome).

The choice of dsRNA as stress trigger allowed us to recapitulate the PKR-mediated ISR activation and translation repression in CEODs. We identified a threshold of 50% residual protein synthesis activity below which G3BP1 phase separation occurs in our system. Although the threshold difference observed between Huh7 and U2OS cells appears to be primarily influenced by variations in PKR expression levels, we anticipate that the levels of G3BP1 and other SG core proteins, in particular Caprin1 or the negative regulator USP10[19], may also modulate the system. Thus, the origin and molecular composition of cell lines should be carefully considered when examining stress-induced LLPS.

Our findings suggest that 200-nt long ssRNA as negative control for PKR activation does not act as a molecular crowder in CEODs under the tested conditions. Although several studies have described that ssRNA can act as a SG scaffold in vitro[48,91–94], three principal differences can support this observation. First, the threshold at which G3BP1 phase separates in vitro is considerably higher (2.5 to 12.5 μM)[21,52] than the concentration we estimated in U2OS YFP-G3BP1 cells. Second, the length of the ssRNA we used is shorter than the minimal length of 250 nt ssRNA demonstrated to promote G3BP1 LLPS[21], which is also in accordance with the evidence that longer mRNAs are recruited to SGs[38]. Additionally, in CEs, the presence of RBPs will significantly restrict RNA-RNA interactions and RNA-driven G3BP1 LLPS, as previously demonstrated in vitro for the RNA helicase DDX19A, which decreases the ability of RNA to promote G3BP1 LLPS[21].

The CEOD system appears to be less responsive than cells with regard to the concentration of drugs to be used. This was observed for the treatment with silvestrol, which induced SGs at a concentration of 0.5 μg/ml *in cellulo*, but required 2 μg/ml to induce YFP-G3BP1 condensates in CEODs. Despite these limitations, the use of the USP10-derived peptide demonstrated that this system offers significant competitive advantages in the screening of SG inhibitory molecules that are not available in a cell membrane-permeable form or are toxic.

The CEOD approach has several technical limitations. For instance, CE preparation may pre-assemble mRNP seeds; pelleting could co-enrich cellular debris with similar sedimentation properties, induce condensate solidification, inflating false positives (as suggested by the large size of the detected proteins); and washing steps may lose weakly interacting SG clients lacking strong RNA or granule associations. To address potential false positives and identify condition-specific differences from the initial protein lists, it was essential to further prioritise hits using (i) unsupervised hierarchical clustering across all conditions and (ii) differential abundance analysis. Together, these analyses trimmed the lists of specific proteins and revealed clear, biologically relevant differences in condensate composition, distinguishing condensates induced by translation repression (dsRNA, silvestrol) from condensates induced through non-physiological molecular crowding (PEG). Interestingly, these analyses also revealed that PEG treatment enriches proteins with high abundance in CEs while dsRNA treatment triggers the selective condensation of low-abundant proteins, often undetectable in the input CEs. In addition, most high-confidence SG components are specifically and non-randomly enriched in dsRNA- and silvestrol-induced condensates.

Although canonical SG-associated terms such as RNA binding and translation-related processes were identified among the significantly enriched biological processes, these terms did not exhibit the highest enrichment scores in our dataset. In fact, proteins enriched in dsRNA-induced condensates predominantly belong to mitochondria-related GOBPs. Among these, we identified four mitochondria-related proteins—MGME1, MRRF, NDUFS6 and MRPL51—that are selectively enriched in dsRNA-induced condensates in CEODs. We confirmed the selective enrichment of MGME1, an exonuclease involved in mitochondrial genome maintenance, in cellular SGs induced with dsRNA, but not with silvestrol or arsenite. This result supports a potential functional link between mitochondria and SG biology, consistent with previous reports[95]. For example, SGs formed under metabolic stress modulate mitochondrial function and fatty acid permeability. Their interactome includes mitochondria-associated proteins, indicating a direct role in metabolic adaptation through interactions with mitochondria[96]. The close proximity of SGs with mitochondria has been observed in heat shock conditions[28], as well as during chronic infection with hepatitis C virus, a positive-sense ssRNA virus that produces dsRNA replication intermediates[45]. In fact, infections with RNA viruses frequently induce mitochondrial remodelling[97]. Collectively, these findings support the idea that mitochondria may play an important role in the formation or function of SGs during dsRNA-induced stress. Comparative proteomic analyses of dsRNA-induced condensates and virus-induced SGs in future studies may provide deeper insights into the common and distinct mechanisms regulating SG biology during antiviral responses.

Importantly, this study identified four E3 ligases as SG client proteins. The ubiquitin machinery is involved in the effective clearance of SGs by the proteasome and by autophagy[98–100]. Among these are histone E3 ligase 2 (Hel2)[101], anaphase-promoting complex[102], and TRIM21[103]. Free ubiquitin has also been detected in SGs[104]. However, different stressors affect differently the mRNPs' ubiquitinome[99,100]. Noteworthy, all E3 ligases identified in this study have been previous associated with human diseases and inflammatory processes. FBXO3 has been identified as a pro-inflammatory factor involved in the inflammasome regulation, cytokine release and neuroinflammation[105–107]. It has also been linked to amyotrophic lateral sclerosis (ALS), a neurological disorder affecting motor neurons, where it fails to promote the degradation of the ALS-mutant form of C21ORF2, leading to an aberrant neurite length in mice[108]. TRIM37 was recently identified in nonhuman primates as a regulator of the degradation of the Huntingtin mutant form, the protein causative of Huntington disease[109]. TRIM38 has been implicated in numerous innate immune and inflammatory pathways, acting as both a positive and negative regulator. Its SUMO E3 ligase activity was found to

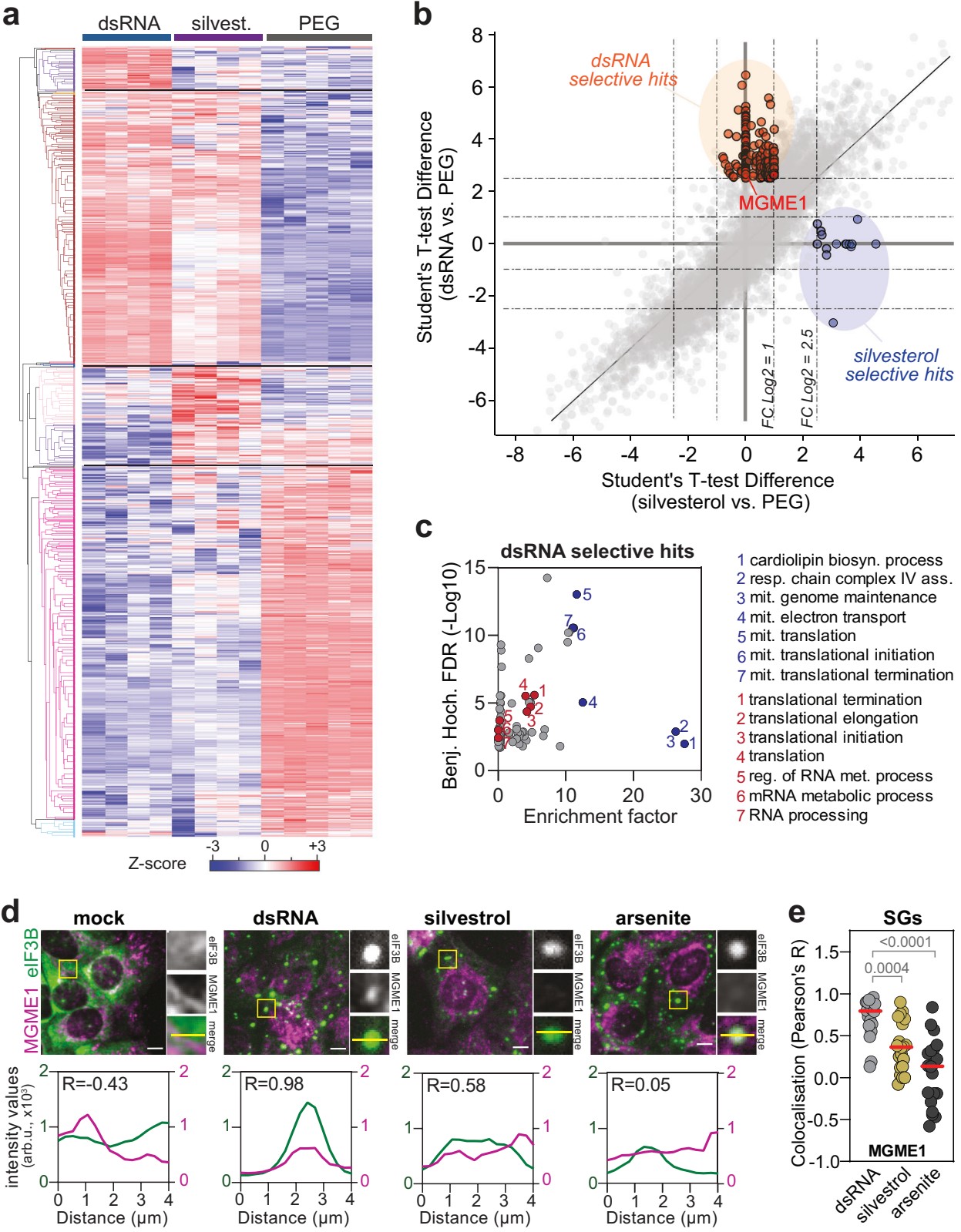

**c** **dsRNA selective hits**

1 cardiolipin biosyn. process
2 resp. chain complex IV ass.
3 mit. genome maintenance
4 mit. electron transport
5 mit. translation
6 mit. translational initiation
7 mit. translational termination

1 translational termination
2 translational elongation
3 translational initiation
4 translation
5 reg. of RNA met. process
6 mRNA metabolic process
7 RNA processing

positively regulate the cytosolic RNA sensor molecules RIG-I and MDA5, as well as the subsequent activation of interferon-stimulated genes[110]. In contrast, its ubiquitination activity was shown to negatively regulate the TLR3-mediated induction of interferons[111] and to suppress the interferon pathway activation by modulating RIG-I levels in a negative feedback loop[112]. Finally, HERC3 has been shown to facilitate

the selective clearance of the misfolded cystic fibrosis transmembrane conductance regulator, a transporter protein dysregulated in cystic fibrosis[113]. The association of these newly identified SG-resident E3 ligases with inflammation in neurodegenerative diseases warrants further investigation, especially with respect to their involvement in SG assembly or disassembly processes.

**Fig. 9 | YFP-G3BP1 condensate proteome.** Condensates were induced by treatment with dsRNA ($n = 4$ technical replicates), silvestrol ($n = 4$ technical replicates), or PEG ($n = 5$ technical replicates), pelleted and analysed by LC-MS/MS. **a** Unsupervised hierarchical clustering of the z-scored profiles of the 6060 significantly changed proteins across treatment groups (multiple-sample ANOVA, S0 = 0, permutation-based FDR < 0.05). **b** Identification of dsRNA exclusive hits (Student's $t$-test difference for dsRNA vs PEG fold change > 2.5 and Student's $t$-test difference for silvestrol vs PEG fold change <1) and of silvestrol exclusive hits (Student's $T$-test difference for silvestrol vs PEG fold change > 2.5 and Student's $t$-test difference for dsRNA vs PEG fold change <1). **c** Enriched gene ontology biological processes (GOBP) annotations for dsRNA exclusive hits. Coloured dots represent the 7 most enriched GOBP terms (blue) and 7 GOBP terms commonly identified in the RNA Granule Database (SG tier 1) (red) (one-sided Fisher's exact test, Benjamini–Hochberg FDR < 0.02). **d, e** MGME1 localisation in U2OS cells transfected with dsRNA, treated with silvestrol, or with arsenite. Untreated cells (mock) served as control. **d** Shown are representative images for each condition and cropped sections of SGs (yellow squares). Scale bar, 5 μm. eIF3B was used as a SG bona fide marker. Intensity profiles for eIF3B and MGME1 fluorescence signals are shown at the bottom. Colocalisation was assessed using Pearson's correlation coefficient (R), which quantifies the linear relationship between pixel intensities in the two fluorescence channels along the line indicated in the cropped section. **e** Scatter plot shows Pearson's R correlation coefficient for 20 SGs in different cells treated with dsRNA (grey) and silvestrol (brown), or arsenite (grey). Statistical significance (1-way ANOVA, Dunnett's multiple comparisons test) compared to dsRNA is indicated.

Furthermore, we identified DNAJC6, also named PARK19 or Auxilin in neurones, a chaperone protein of the HSP40 family. A homozygous mutation in the gene *DNAJC6*, which results in severely reduced levels of Auxilin, has been associated with juvenile parkinsonism, another neurodegenerative disease[114,115], and synaptic dysfunctions[116]. HSPs, including HSP27, HSP70 and HSP90, play an important role in the dynamics, especially the disassembly, of SGs[117]. Chaperones such as HSPB8 and HSP70 regulate the composition and dynamics of SGs, preventing the accumulation of aberrant SGs, and thereby playing a potential role in the development of ALS disease[118,119].

Glycosylation is an important PTM regulating SG dynamics[77]. A variety of stress stimuli have been shown to promote the reversible O-GlcNAcylation modifications of cellular proteins, including some required for SG assembly[120]. Indeed, silencing of the associated genes O-β-N-acetylglucosaminyltransferase (OGT), sortilin (SORT1), and glutamine:fructose-6-phosphate amidotransferase 2 (GFAT2) strongly reduced SG assembly in response to arsenite[121]. Conversely, O-GlcNAcylation has also been involved in heat shock-induced SG disassembly[122]. In this instance, the key translation initiation factor eIF4GI is O-GlcNAcylated, which results in the repulsion of poly(A)-binding protein 1 (PABP1) and promotes SG disassembly, facilitating selective translation of stress mRNAs[122]. Here, we identified MGAT5, a major player in the N-GlcNAcylation branch of the glycosylation pathway. Interestingly, MGAT5 expression has been previously associated with inflammation, chronic viral infection and CD8 + T cell responses, through addition of N-acetylglucosamine on surface glycoproteins[123]. Although the importance of N-GlcNAcylation for SG dynamics requires further investigation, selective accumulation in SG might be a potential mechanism allowing finely regulated responses to selected cellular stressors.

SG disassembly is a tightly regulated, multistep process involving numerous disassembly-engaged proteins[26,54]. Several proteins that coordinate ubiquitin and small ubiquitin-like modifier (SUMO) conjugation signaling pathways, including SUMO conjugation enzymes, the HECT domain E3 ligases ITCH and NEDD4L, and the ubiquitin receptor TOLLIP[54,78], localise to SGs and contribute to their resolution. In this study, we identified FBXO3 as a modulator of ITCH expression, supporting a model in which balanced E3 ligase activity temporally regulates SG persistence. This precise timing of SG clearance may provide a critical window for appropriate cellular responses.

Collectively, we have developed a versatile, cost-effective, near-physiological CEOD system that offers a robust and adaptable platform for elucidating the mechanisms underlying stress responses and related diseases, while also facilitating the identification of potential therapeutic targets.

## Methods
### Cell lines
U2OS and U2OS ΔPKR[36], U2OS ΔΔG3BP1/2 (ΔΔG1/2) and U2OS ΔΔG3BP1/2 mCherry-G3BP1 (ΔΔG1/2 mChG1)[19] cells, Huh7 cells and HEK293T cells were maintained at 37 °C and 5% $CO_2$ in Dulbecco's modified Eagle´s medium supplemented with 10% FCS (Capricorn Scientific), 1% non-essential amino acids, 100 U/ml penicillin and 100 μg/ml streptomycin (all from Gibco, Life Technologies). Stables cell lines stably expressing YFP-TIA1 or YFP-G3BP1 were maintained with additionally 1 mg/ml G418.

### Plasmids
The following plasmids have been described previously: pWPI YFP-TIA1 Neo, lentiviral vector encoding for YFP-TIA1 and the neomycin resistance gene[124]; pTM1.2 eGFP, a plasmid containing the T7 RNA polymerase promoter followed by the eGFP gene under the control of the internal ribosome entry site from encephalomyocarditis virus[125].

Luciferase reporters: pGL3-β-Globin 5′UTR-Firefly, encoding FLuc under the control of the rabbit β-globin 5′ UTR: the sequence of T7 RNA polymerase promoter, followed by the β-globin 5′ UTR sequence, flanked on the 5′ and 3′ ends by HindIII and Nco I restriction sites, respectively was synthesized by GeneArt (Thermo Fisher Scientific) and inserted upstream of the Fluc sequence into pGL3 basic (Promega) using the same restriction sites. The plasmid pGL3-β-Globin 5′UTR-Renilla was generated by replacing the FLuc sequence in the pGL3-β-Globin 5′UTR-Firefly vector by that of the *Renilla* luciferase using an overlapping PCR strategy.

Lentiviral vectors: pWPI YFP-G3BP1 Neo was generated in two steps. First, the eYFP sequence was excised from peYFP-TIA1 (kindly provided by P. Anderson) using the restrictions enzymes EcoRI and BglII (New England Biolabs, NEB). The fragment was inserted into pCL puro GFP-G3BP1 (kindly provided by G. Stoecklin), digested with the same restriction enzymes to generate the plasmid peYFP-G3BP1. Second, the YFP-G3BP1 sequence was excised by digesting peYFP-G3BP1 with AgeI and EcoRI. The fragment was blunted and inserted into the lentiviral vector pWPI Neo linearised with PmeI. Note that we identified a point mutation within its stop codon in the G3BP1 originating plasmid, which led to the additional expression of 16 additional amino acids at the C-terminal end of G3BP1. A corrected YFP-G3BP1 expressing plasmid was generated for the generation of new stable cell lines.

### Generation of cell lines expressing stably YFP-G3BP1 and YFP-TIA1
Stables cell lines were generated by lentiviral transduction as previously described[124]. In brief, HEK293T cells ($5 \times 10^6$) were transfected with 6.42 μg pWPI eYFP-G3BP1 Neo or pWPI YFP-TIA1 Neo, 6.42 μg packaging plasmid pCMV Δ8.91 and 2.16 μg VSV-G encoding plasmid pMD2.G (kind gifts from Didier Trono). DNA was resuspended in Opti-MEM (Gibco, Life Technologies), mixed with polyethylenimine in a 1:3 (PEI:DNA) ratio, incubated for 20 min at room temperature, and added dropwise to the cells for 6 h. Lentivirus-containing supernatants were harvested after 48 h, filtered through a 0.45 μm pore-sized membrane and used to transduce target cells. Transduced cells were selected in the presence of 1 mg/ml G418 for three days. Higher YFP-G3BP1- and YFP-TIA1-expressing cells were sorted by fluorescence-activated cell sorting. Note that stable Huh7 and U2OS cells lines expressing YFP-G3BP1 with the corrected original stop codon were similarly generated

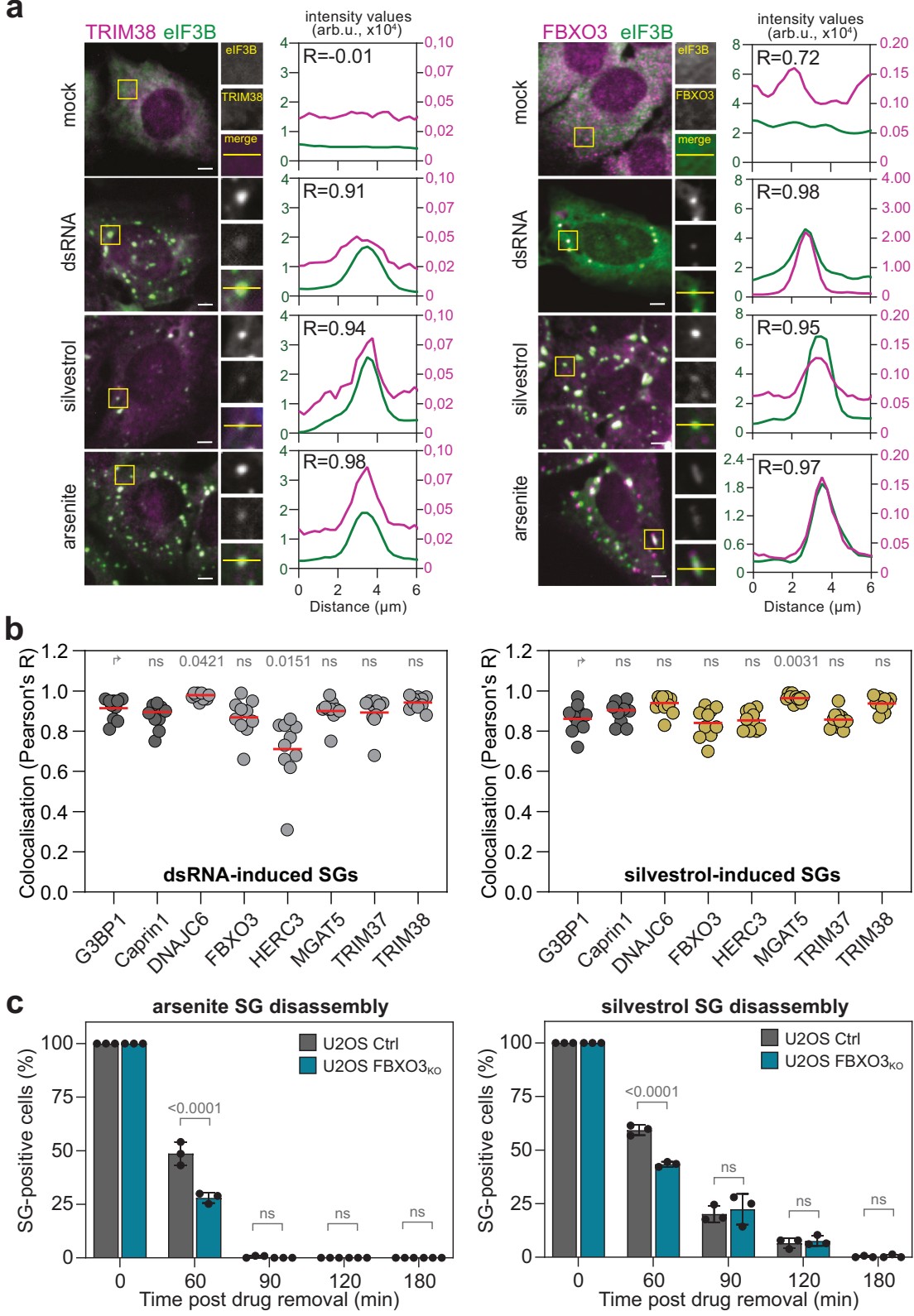

and phenotypically compared to the previous YFP-G3BP1 expressing lines. Supplementary Fig. 21 demonstrates that comparable results were obtained for all key findings.

### Generation of knockout cell clones

Huh7 PKR$_{KO}$ and U2OS FBXO3$_{KO}$ single cell clones were generated using a CRISPR/Cas9 two-guide deletion approach as previously

reported[126]. Predesigned crRNAs targeting *EIF2AK2* exon 3 (Hs.Cas9.EIF2AK2.1.AC: 5′−/AltR1/TTATCCATGGGGAATTACATG/AltR2/−3′) and exon 5 (Hs.Cas9.EIF2AK2.1.AB: 5′−/AltR1/TAATACATACCGTCA-GAAGCG/AltR2/−3′) and targeting FBXO3 exon 2 (Hs.Cas9.FBXO3.1.AA: 5′−/AltR1/TATCAAGTCATGATCCGCTG/AltR2/−3′) and exon 3 (Hs.Cas9.FBXO3.1.AC: 5′−/AltR1/TAAAACCATCCGAGGACACC/AltR2/−3′) were purchased from Integrated DNA Technologies (IDT).

**Fig. 10 | Identification of SG client proteins.** The localisation of candidates identified by LC-MS/MS in SGs was validated by immunofluorescence analysis in U2OS cells transfected with dsRNA, treated with silvestrol, with arsenite or left untreated (mock). **a** Shown are representative images and cropped sections of SGs (yellow square) for TRIM38 (left panel) and FBXO3 (right panel) localisation. Scale bar 5 μm. eIF3B was used as a SG marker. Intensity profiles for eIF3B and TRIM38 or FBXO3 fluorescence signals are shown on the right. Colocalisation was assessed using Pearson's correlation coefficient ($R$), which quantifies the linear relationship between pixel intensities in the two fluorescence channels along the line indicated in the cropped section. **b** Colocalisation of the newly identified SG proteins with dsRNA- (left panel) or silvestrol-induced (right panel) SGs. Scatter plots show

Pearson's R correlation coefficient for 10 SGs in different cells. Statistical analysis (1-way ANOVA, Tukey's multiple comparisons test) compared to G3BP1 is indicated. ns non-significant ($p \geq 0.05$). **c** Role of FBXO3 in SG disassembly. U2OS control cells (Ctrl) and FBXO3$_{KO}$ cells were treated with 0.5 mM arsenite (left panel) or 2 μg/ml silvestrol (right panel). After treatment, drugs were removed, cells washed with PBS, and medium replenished (0 min) to allow SG disassembly. Cells were fixed and analysed by fluorescence microscopy at the indicated timepoints ($n = 3$ independent biological replicates). Values (mean ± SD) represent percentage of SG-positive cells. Statistical analysis (2-way ANOVA, Tukey's multiple comparisons test) compared to Ctrl cells is indicated. ns non-significant ($p \geq 0.05$).

Negative control clones were generated using a predesigned non-targeting crRNA (IDT), following the same protocol. In brief, individual crRNAs and tracrRNA were mixed in equimolar amounts and hybridised for 5 min at 95 °C, followed by stepwise cooling to room temperature. Pre-assembled crRNA/Cas9 ribonucleoprotein complexes (RNPs) were obtained by gently mixing 1.2 μl of hybridised RNAs with 17 μg of recombinant Cas9 protein (IDT) and 2.1 μl of PBS 1×, followed by 20 min incubation at room temperature. Nucleofection was carried out as recommended by the manufacturer using the Nucleofector Solution T (Lonza) and the pre-set program T-22 of the Amaxa 2b nucleofector device (Lonza) for Huh7 cells and the SE Cell Line Nucleofector Solution (Lonza) with the pre-set program CM-104 for U2OS cells. Nucleofected cells were incubated for 48 h prior to clonal amplification and screening for homozygous clones using target-specific PCR (GoTaq Hot Start, Promega) on genomic DNA using the following primers: PKR_KO_Fwd: 5′–GGA AGA AGA AAT GGC TGG TG–3′ and PKR_KO_Rev: 5′–CCA CCA TCA TGT CCT CAA TG–3′; FBXO3_KO-Fwd: 5′–TGG TTA CTG CC AGC TG TTG T–3′ and FBXO3_KO-rev: 5′–ATG GAA ACG GGG AAA GGG AC–3′. PCR products were further subcloned into a TOPO TA cloning vector (Thermo Fisher Scientific) according to manufacturer's recommendation and transformed into DH5α competent bacteria. Eight to ten colonies were selected and amplified for the extraction of plasmid DNA, which was then send for sequencing to ensure that all potential allelic sequence variants were covered. In addition, for each single KO cell clone, expression levels FBXO3 were analysed by Western blotting. Expression levels of PKR were analyseed after induction with 1000 IU/ml IFN-α (PBL Assay Science) for 24 h.

## In vitro transcription of reporter mRNAs
In vitro transcription (IVT) reactions were performed as previously described[57]. For each IVT reaction, 10 μg of plasmids were linearised by restriction digestion with XbaI (for pGL3-β-blobin-5′UTR- and pGL3-β-globin-5′UTR-Fluc) or with SpeI (for pTM1.2 eGFP) and purified using the Nucleospin Gel and PCR Clean-Up kit (Macherey-Nagel). Reactions he reaction were performed in a total volume of 100 μl. For capped transcripts (firefly and *Renilla* mRNAs), the linearised DNA was mixed with 10 μl of transcription buffer (80 mM HEPES pH 7.5, 12 mM MgCl$_2$, 2 mM spermidine, and 40 mM dithiothreitol) 12.5 μl of NTP mix (25 mM ATP, UTP and CTP and 12.5 mM GTP), 20 μl of a 5 mM solution of m7G(ppp)G RNA cap analogue (NEB), 2.5 μl of RNAsin (40 U/μl, Promega), and 4 μl of T7 RNA polymerase (home-made). For uncapped mRNA (eGFP), the linearised DNA was mixed with 10 μl of transcription buffer, 12.5 μl of NTP mix (25 mM ATP, UTP, CTP, GTP), 2.5 μl of RNAsin (40 U/μl; Promega), and 4 μl of home-made T7 RNA polymerase. IVT reactions were incubated for 4 h at 37 °C. Next, DNA template was digested by addition of 10 μl RQ1 RNase-free DNase (1 U/μl, Promega) for 30 min at 37 °C. Transcripts were purified by phenol/chloroform/isoamyl alcohol (25/24/1 v/v, Applichem) extraction and isopropanol precipitation. RNA pellets were resuspended in RNase-free water. Purified firefly and *Renilla* reporter mRNAs were further poly-adenylated using the *E. coli* Poly(A) polymerase (NEB). In brief, 5 μg of transcripts were mixed with 10 mM ATP, *E. coli* Poly(A) Polymerase

Reaction Buffer and 5U *E.coli* Poly(A) in a final volume of 20 μl, and incubated for 30 min at 37 °C. The polyadenylated mRNAs were further purified using the Monarch RNA Clean Up Kit (NEB) and RNA eluted in nuclease-free water. RNA integrity was determined by agarose gel electrophoresis and measurement of ratio of absorption at 260 and 280 nm.

## Production of translation-competent cytosolic extracts (CEs)
Translation competent cellular extracts were produced as described elsewhere[127]. In brief, cells were grown in 15-cm cell culture dishes to a confluence of 90% (approx. $7 \times 10^6$ cells). Cells were washed with PBS, detached by trypsinisation, resuspended and pelleted at $4000 \times g$ for 5 min. The cell pellet was resuspended with an equivalent volume of ice-cold hypotonic extraction buffer (10 mM HEPES, pH 7.6, 10 mM potassium acetate, 0.5 mM magnesium acetate, 5 mM DTT, supplemented with protease inhibitors (cOmplete, EDTA-free protease inhibitor cocktail, Roche). After swelling on ice for 45 min, cells were lysed by mechanical disruption using a 27 G needle, and pelleted at $14,000 \times g$ for 1 min. Cell extract protein concentration was measured using a Bradford assay (Protein Assay, Bio-Rad) and adjusted to 20 μg/μl before storage at −80 °C.

## In vitro translation reactions
In vitro translation reactions in bulk were carried out as described elsewhere[127] in a final volume of 15 μl, with a CE final concentration of 5 μg/μl unless stated otherwise. Seventy-five μg of cellular extract was mixed with 1.5 μl translation buffer (16 mM HEPES pH 7.6, 20 mM creatine phosphate, 0.1 μg/μl creatine kinase, and 0.1 mM spermidine, 100 μM L-amino acids) supplemented with 150 mM KAc, 2.5 mM MgAc, 10 U RNasin ribonuclease inhibitor (Promega) and 100 ng luciferase-encoding transcript or 500 ng eGFP-encoding transcript. Reactions were carried at 37 °C for 2 h, unless differently indicated and stopped by addition of 200 μg/ml cycloheximide (CHX, Sigma-Aldrich). Control reactions were carried in presence of 200 μg/ml CHX, or 10 nM GMP-PNP (Sigma Aldrich). Translation repression was induced by addition of the indicated concentrations of 200-bp dsRNA[57], PEG-8000 (AppliChem), silvestrol (Biozol), or sodium arsenite (Sigma-Aldrich) to the reaction mix before incubation for 2 h at 37 °C. For translation in CEODs, the translation reaction mix was prepared on ice and immediately encapsulated in surfactant-stabilised oil before incubation at 37 °C.

## Luciferase activity measurements
Luciferase activity was measured in duplicate using a tube luminometer (Lumat LB 9510, Berthold Technologies) from at least 3 biological repeats. For the measurements of FLuc activity, half of the translation reaction was mixed with 100 μl of freshly prepared luciferase assay buffer (25 mM Glycyl-Glycine pH 7.8, 15 mM MgSO$_4$, 15 mM K$_2$PO$_4$, and 4 mM EGTA) supplemented with 200 μM D-Luciferin (PJK biotech), 2 mM ATP and 1 mM DTT. Luminescence signal was measured for 20 s. For detection of *Renilla* luciferase activity, half of the translation reaction was mixed with 50 μl luciferase assay buffer supplemented with 14 nM coelanterazine (PJK Biotech). Luminescence

signal was measured for 10 s. The luciferase activity served as surrogate for cell extract translation levels and was displayed as relative light units (RLU) or as normalised value to the control.

Pleotrophic effects of arsenite treatment were assessed by adding 1 ng/ml recombinant FLuc (Sigma-Aldrich) to U2OS CEs in presence of 0.5 mM arsenite. Firefly luminescence was measured after 2 h incubation at 37 °C as described above.

### Induction of translation repression and phase separation in CEs
Unless indicated differently, translational repression and phase separation were induced by supplementing the translation reaction mix with 250 nM dsRNA[57], 2 µg/ml silvestrol (Biozol), 1% PEG-8000 (AppliChem) or 1 µM ISRIB (Sigma-Aldrich). Appropriate controls included dimethyl sulfoxide (DMSO, Merck Millipore), 250 nM ssRNA and 250 nM dsDNA with matching sequence to that of the 200-bp dsRNA. Recombinant mScarlet (25 mM, home-made) or recombinant cyan fluorescent protein (25 mM, Antibodies-online GmbH) were additionally spiked into the reaction mix for visualisation of the CEODs by microscopy. Reaction mixes were encapsulated and incubated at 37 °C for 2 h, unless indicated differently.

### Surfactant-stabilised cytosolic extract-in-oil droplets (CEODs)
For the formation of CEODs was adapted from[60]. The translation reaction (15 µl) was layered on 45 µl fluorinated oil (Novec 7500, >99%, 3 M) containing 1.4% PEG-based fluorosurfactant (008-FluoroSurfactant-1G, Ran Biotechnologies) (1:3 v/v). The phases were mixed by gently scraping tube surface onto the surface of the Eppendorf tube rack and kept on ice before incubation for 2 h at 37 °C. Ten µl of emulsion containing the CEODs was loaded in a sealed home-made glass observation chamber, made by joining a glass slide and a cover slip with double-sided tape, then sealing the edges with nail polish. When necessary, CEODs were destabilised to release their content by addition of 3 volumes of 1H,1H,2H,2H-Perfluor-1-octanol (Sigma-Aldrich) to the emulsion and gentle rotation of the tube until separation of two phases. The upper phase containing CE was carefully transferred into a new tube for further analysis.

### Micrococcal nuclease treatment of CEs
Cellular RNAs were digested using *Staphylococcus aureus* micrococcal nuclease (NEB). In brief, 80 µg CEs were treated with 18 U micrococcal nuclease and 0.7 mM $CaCl_2$ in a final volume of 6.4 µl for 10 min at 25 °C. The reaction was stopped by addition of 2.5 mM EDTA. Nuclease-treated extracts were immediately used for in vitro translation assays.

### Analysis of CEODs by spinning disc microscopy
The presence of condensates was analysed by spinning disc microscopy using a Nikon Eclipse Ti2 with a Confocal Scanning Unit (Crest X-Light V3) and a Zyla sCMOS camera (Andor) at room temperature unless specified otherwise. 2D images were acquired using a Nikon Plan Apo VC 20× (NA 0.75) objective. We used a dry lens, as immersion objectives, although having a higher resolution, create a thermal bridge with the sample, leading to convective movement of CEODs and condensates within them. The equatorial plane of the CEODs was identified and analysis performed in 2D to quantify extend of condensation.

For real-time imaging of dsRNA-induced phase separation, U2OS mCherry-G3BP1 YFP-TIA1 CEs were mixed with 250 nM dsRNA and encapsulated. Directly afterwards, CEODs were transferred to an observation chamber kept on ice. The observation chamber then was transferred into the pre-heated stage top incubation chamber (Oko-Lab). The fluorescence signal of mCherry-G3BP1 and YFP-TIA1 in CEODs were imaged with an interval of 2 min for 2 h at 37 °C.

### Condensate analysis pipeline
An automated image analysis pipeline was created in Nis-Elements 5.21 using the GA3 tool. First, CEOD detection was performed using the mScarlet signal of the RFP channel. The RFP channel was pre-processed with a Gaussian filter to remove noise. A manual intensity threshold was set to segment the CEODs. This threshold was manually adjusted for each set of experiments. Second, condensate detection was performed using the YFP-G3BP1 signal, which was pre-processed with the functions "denoise AI" and "rolling ball correction" to remove background. Condensates were segmented using an intensity threshold of 60 to allow comparability between conditions. Segmented condensates were then saved as binary images and attributed to the segmented CEOD in which they were located using the "mother-child" function. Morphological features such as condensate area (µm²), perimeter (µm) and circularity were extracted from the segmented condensates. The circularity was calculated using the following formula: $4\pi*area/perimeter^2$. Circularity values range between 0 and 1, with 1 corresponding to a perfect circle). Additionally, the percentage of CEOD area covered by condensate was calculated by dividing the condensate area by the CEOD area.

### G3BP1 mobility in CEODs
G3BP1 mobility analyses were performed using U2OS YFP-G3BP1 CEs and CEODs (Supplementary Fig. 1g, h), U2OSΔΔG1/2 mCh-G3BP1 CEODs (Fig. 8a, b). FRAP experiments were performed using a PerkinElmer UltraVIEW VoX spinning disc confocal microscope (Nikon Eclipse Ti-E) equipped with CSU-X1 spinning disk unit (Yokogawa). Images were acquired using a 40× Plan APO lambda air (NA 0.75) objective (Nikon) and a sCMOS ORCA-Flash 4.0 camera (2048 × 2048 px, Hamamatsu Photonics). The region of interest was manually selected and photobleached at 100% laser intensity (561 nm, 50 mW) for one second using a beam radius of 10 µm. Bleaching and fluorescence recovery were monitored at 500 ms interval for 3 min. Images were analysed using the image processing package Fiji (http://fiji.sc/wiki/index.php/Fiji)[128]. The mean signal intensity of YFP-G3BP1 and mCherry-G3BP1 was measured in the bleached area and the value normalised to the background intensity in a non-bleached region of the same CE or CEOD. Images were analysed using the image processing package Fiji. The YFP-G3BP1 and mCherry-G3BP1 mean signal intensity was measured in the bleached area and the value normalised to the background intensity in a non-bleached region of the same CE or CEOD. The half-maximal fluorescence intensity was determined by fitting a non-linear regression curve.

### Induction of SGs in cells
For immunofluorescence analysis, cells ($3 \times 10^5$ cells/well) were seeded on a coverslips in a 12-well plate. For dsRNA transfection, 1 µg dsRNA was diluted in Opti-MEM (Gibco) in a 1:3 ratio with Lipofectamine 2000 (Invitrogen) as recommended by the manufacturer, incubated for 20 min at room temperature and added dropwise to cells for 16 h. Additionally, cells were treated with with 2 µg/ml silvestrol for 1 h at 37 °C or with 0.5 mM arsenite (Sigma-Aldrich) for 45 min. After treatment, cells were fixed for 15 min with 4% PFA in PBS and coverslips stored for immunofluorescence analyses. For protein expression analysis, cells ($2 \times 10^6$) were seeded in a 6-cm dish, treated similarly and lysed for further Western blot analysis.

### Western blot analysis
Cells were scraped in ice-cold protein lysis buffer (50 mM Tris HCl pH 7.4, 150 mM NaCl, 1% Triton ×-100, 60 mM β-glycerophosphate, 15 mM 4-nitrophenylphosphate, 1 mM $Na_3VO_4$, 1 mM NaF) supplemented with cOmplete EDTA-free protease inhibitor cocktail (Roche) and incubated on ice for 30 min before pelleting at $16,200 \times g$ for 30 min at 4 °C. Proteins were denatured and separated on a 10% polyacrylamide gel (20 µg for the analysis of eIF3b, PKR and eIF2α; 50 µg for the analysis of the phosphorylated forms). Electrophoresis was conducted in 1× TGS buffer (25 mM Tris pH 8.6, 192 mM glycine, 0.1% SDS) at 80–120 V and protein was transferred to a methanol pre-activated polyvinylidene

difluoride membranes (PVDF 0.45 µm pore-sized, Merck Millipore) in wet-blot transfer buffer (2.5 mM Tris, 15 mM Glycine, pH 8.3, 20% methanol). Membranes were stained with Ponceau S solution (Sigma-Aldrich), scanned and destained with TBS-T buffer (25 mM Tris-HCl, 150 mM NaCl, 2 mM KCl, pH 7.4, 0.1% Tween). Depending on antibody requirements, membranes were blocked with 5% protease-free BSA (Sigma-Aldrich) or 5% non-fat dry milk (Roth) in TBS-T buffer. The following primary antibodies were used at indicated dilutions and membranes incubated overnight at 4 °C: p-eIF2α (1:1000; BSA; Cell Signaling Technology, CST), eIF2α (1:1000; milk; CST), PKR (1:1000; BSA, Proteintech), p-PKR (1:500; BSA; Abcam), eIF3b (1:5000; BSA; Bethyl Laboratories), GFP (1:1000 BSA; Roche), G3BP1 (1:1000; BSA; Bethyl Laboratories). FBXO3 (1:500; milk; Santa Cruz Biotechnology); NEDD4L (1:2000; milk; Proteintech), WWP2 (1:2000; milk; Proteintech), ITCH (1:2000; milk; Proteintech), Akt (1:1000; BSA; CST), p-Akt (1:1000; BSA; R&D Systems), HSP27 (1:1000; BSA; CST), p-HSP27 (1:1000; BSA; CST), p38 (1:1000; BSA;), p-p38 (1:1000; BSA; CST). Membranes were washed 3 times with TBS-T buffer prior to addition of horseradish peroxidase-conjugated secondary antibodies diluted in blocking solution (anti-mouse 1:10,000 and anti-rabbit 1:10,000 from Sigma-Aldrich) for 1 h at room temperature. Membranes were washed 3 times in TBS-T buffer. Chemiluminescence detection was performed using Western Lightning Chemiluminescent Substrate (Perkin Elmer) according to the manufacturer's instructions. Chemiluminescence signals were detected with the ChemoCam imager (INTAS Science Imaging Instruments). Band signal intensity was quantified using the LabImage1D software (INTAS Science Imaging Instruments), normalised to the indicated loading controls as shown as arbitrary units (arb.u.) or fold control. Uncropped and unprocessed scans of the main figures are provided in the Source Data files. Those of Supplementary Figures are provided in the Supplementary Information.

### Protein solubility assay

U2OS CEs (100 µg) were treated with 250 nM dsRNA, 2 µg/ml silvestrol, DMSO or untreated (mock) for 2 h at 37 °C. After incubation, samples were centrifuged for 10 min at $14{,}000 \times g$ at 4 °C. Pellets were washed twice with 20 µl ice cold PBS and resuspended in 40 µl of RIPA buffer (50 mM Tris-HCl pH 7.4, 150 mM NaCl, 1% Triton ×-100, 1 mM EDTA, 1 mM NaF, 200 nM Na₃VO₄, 0.1% SDS, Complete Protease Inhibitor Cocktail (Roche)). Samples were centrifuged at $14{,}000 \times g$ for 20 min at 4 °C to separate the "RIPA soluble fraction" from the detergent-insoluble pellet. Pellets were further washed twice with 20 µl of ice-cold PBS and resuspended in 20 µl urea buffer (8 M urea, 5% SDS, in 1× RIPA buffer) ("urea soluble fraction"). Samples of the "RIPA soluble fraction" were mixed with 6× protein sample buffer (375 mM Tris-HCl pH 6.8, 50% glycerol, 9% SDS, 0.03% bromophenol blue) supplemented with 1.5% β-mercaptoethanol final and denatured by heating at 95 °C for 5 min. Samples of the "urea soluble fraction" were mixed with 6x protein sample buffer without boiling. Proteins were resolved by SDS–PAGE and analysed by Western blotting as described above. Protein levels were normalised to total protein as determined by staining with Ponceau S.

### Immunofluorescence analysis

Cells were permeabilised for 5 min with 0.5% Triton-X100 in PBS, washed with PBS, and incubated for 30 min in blocking buffer (5% horse serum, 5% sucrose in PBS). Cells were incubated with primary antibodies diluted in blocking buffer for 1 h: eIF3η (1:500, Santa Cruz Biotechnology), eIF3B (1:500, Bethyl Laboratories), G3BP1 (1:500, Bethyl Laboratories), G3BP1 (1:500, BD Biosciences), Caprin1 (1:500, Proteintech), DNAJC6 (1:50, Abcam), FBXO3 (1:50, Santa Cruz Biotechnology), HERC3 (1:50, Santa Cruz Biotechnology), MGAT5 (1:50, Invitrogen), MGME1 (1:500, Proteintech), TRIM37 (1:50, Santa Cruz Biotechnology), TRIM38 (1:50, Invitrogen). Cells were washed 3 times with PBS for 5 min and incubated with fluorescently labelled secondary

antibodies diluted 1:2000 in blocking buffer for 1 h: donkey anti-mouse Alexa Fluor 488 (Invitrogen), donkey anti-rabbit Alexa Fluor 568 (Invitrogen). Cells were washed 3 times with PBS and once with ddH₂O, before mounting of the coverslips on glass slides using Fluoromount-G DAPI (SouthernBiotech). Fluorescence images were acquired on a Nikon Eclipse Ti-E inverted microscope using a Nikon CFI Plan Apo lambda air 20× objective (NA 0.95) or a Nikon CFI Plan Fluor 40× objective with oil immersion (NA 1.30) and the Nikon NIS-Elements "Advanced Research" software (version 5.30.06). For colocalisation analyses, signal intensity histograms and Pearson's correlation coefficients were calculated on single plane images using the integrated function Coloc2 in Fiji[128].

### Antibody stain in CEODs

Antibody stain in CEODs was performed adapted from[52]. Primary and secondary antibodies were pre-conjugated in translation buffer for 2 h at room temperature with gentle shaking. Primary antibodies (G3BP1; BD Biosciences; Caprin1 Proteintech; DNAJC6, Abcam; FBXO3, Santa Cruz Biotechnology; HERC3, Santa Cruz Biotechnology; NDUFS6, Abcam; MGAT5, Invitrogen; MGME1, Proteintech; MRPL51, Abcam; MRRF, Proteintech; TRIM37, Santa Cruz Biotechnology; TRIM38, Invitrogen) were coupled secondary antibodies labelled to Alexa Fluor 647 for 5 min prior mixing with the YFP-G3BP1 CEs and encapsulation. Imaging was conducted as described above.

### Colocalisation analysis in condensates and in SGs

For colocalisation analyses, Pearson's correlation coefficient (R) was calculated on single plane images, by using the integrated function "Coloc2" in Fiji[128]. In addition, signal intensity histograms were created by using the function "Plot Profiles". For each candidate, the quantification of the Pearson's correlation coefficient was performed for 10 condensates or for 10 SGs in each condition (mock, dsRNA, silvestrol, arsenite).

### YFP-G3BP1 condensate pelleting and sample preparation for RNA sequencing

U2OS CEs were treated with 250 nM dsRNA, 1% PEG, 2 µg/ml silvestrol or untreated (mock) for 2 h at 37 °C. After incubation, samples were centrifuged for 10 min at $14{,}000 \times g$ at 4 °C (3 biological replicates per condition). Pellets were washed twice with 20 µl ice cold PBS and resuspended in 20 µl RNase-free water. Total RNA was extracted using TRIzol reagent (Thermo Fisher Scientific) according to the manufacturer's instructions. The RNA sequencing and library preparation was performed by Azenta Life Sciences (Leipzig, Germany). In brief, following RNA quality control, mRNAs were enriched using poly(A) selection, fragmented, and reverse-transcribed using random hexamer priming. Second-strand cDNA synthesis, 5′ end-repair, 5′ phosphorylation, and dA-tailing were performed prior to adaptor ligation. Libraries were enriched by PCR and sequenced on an Illumina platform (HiSeq) using a $2 \times 150$ bp paired-end configuration, targeting 30 million reads per sample.

### RNA sequencing data preprocessing and differential expression analysis

Raw RNA-seq read counts were processed and analyzed using the DESeq2 package[129] in R. Genes with low read counts (<10) in at least 2 replicates pro condition were filtered out prior to analysis to reduce noise. Data normalisation was performed using DESeq2's median-of-ratios method to account for differences in sequencing depth and RNA composition across samples. Differentially expressed genes (DEGs) analysis was performed to estimate the enrichment and depletion of the individual mRNAs in condensates (dsRNA, silvestrol, PEG-induced, and mock) against the control input based on fold change (log2). The results of DEG analysis with fold change, Wald test score, p-values, and p-values adjusted for multiple testing using the Benjamini–Hochberg

procedure to control the false discovery rate (FDR) are available in Supplementary Data 2.

### YFP-G3BP1 condensate enrichment and sample preparation for quantitative LC-MS/MS proteomics

U2OS CEs were treated with 250 nM dsRNA, 1% PEG, 2 µg/ml silvestrol or untreated (mock) for 2 h at 37 °C (5 replicates per condition). After incubation, samples were centrifuged for 10 min at 14,000 × $g$ at 4 °C. Pellets were washed twice with 20 µl ice cold PBS and resuspended in 100 µl SDS buffer (4% SDS in 50 mM Tris, pH 8.0) before storage at −80 °C. Pellets were resuspended in 4% SDS in 10 mM Tris-HCl (pH = 7.5). The samples were acetone precipitated twice and afterwards resuspended and denatured in 40 µL U/T buffer (6 M urea/2 M thiourea in 10 mM HEPES, pH 8.0) followed by reduction and alkylation in 10 mM DTT and 55 mM iodoacetamide, and digestion with 1 µg LysC (FUJIFILM Wako Chemicals) and 1 µg trypsin (Promega) in 40 mM ABC buffer (50 mM $NH_4HCO_3$ in water, pH 8.0) overnight at 25 °C, 800 rpm. After digestion, peptides were purified on stage tips with 3 layers of C18 Empore filter discs (3 M) as previously described in ref. [130]. Independent replicates ($n = 5$) of the corresponding CE ("input") were prepared as described above and measured alongside the other samples to assess global protein abundances before condensate enrichment.

### Ultra-high-performance liquid chromatography and trapped ion mobility spectrometry quadrupole time of flight settings

Samples were analysed on a nanoElute (plug-in v.1.1.0.27; Bruker) coupled to a trapped ion mobility spectrometry quadrupole time of flight (timsTOF Pro) (Bruker) equipped with a CaptiveSpray source. Peptides (250 ng) were injected into a Trap cartridge (5 mm × 300 µm, 5 µm C18; Thermo Fisher Scientific) and next separated on a 25 cm × 75 µm analytical column, 1.6 µm C18 beads with a packed emitter tip (IonOpticks). The column temperature was maintained at 50 °C using an integrated column oven (Sonation GmbH). The column was equilibrated using 4 column volumes before loading samples in 100% buffer A (99.9% Milli-Q water, 0.1% formic acid (FA)). Samples were separated at 400 nl min−1 using a linear gradient from 2 to 17% buffer B (99.9% ACN, 0.1% FA) over 60 min before ramping up to 25% (30 min), 37% (10 min) and 95% of buffer B (10 min) and sustained for 10 min (total separation method time, 120 min). The timsTOF Pro was operated in parallel accumulation-serial fragmentation (PASEF) mode using Compass Hystar v.5.0.36.0. Settings were as follows: mass range 100–1700 m/z, 1/K0 start 0.6 V·s/cm²End 1.6 V·s/cm²; ramp time 110.1 ms; lock duty cycle to 100%; capillary voltage 1600 V; dry gas 3 l min−1; dry temperature 180 °C. The PASEF settings were: 10 tandem mass spectrometry (MS) scans (total cycle time, 1.27 s); charge range 0–5; active exclusion for 0.4 min; scheduling target intensity 10,000; intensity threshold 2500; collision-induced dissociation energy 42 eV.

### Raw MS data processing and analysis

Raw MS data were processed with the MaxQuant software v.1.6.17.0 using the built-in label-free quantitation algorithm and Andromeda search engine[131], and further processed with the Perseus software v.1.6.15.0[132], as previously described[133]. Protein tables were filtered to eliminate the identifications from the reverse database and common contaminants. In the subsequent MS data analysis, only proteins identified on the basis of at least one peptide and a minimum of three quantitation events in at least one experimental group were considered. The iBAQ protein intensity values were log2-transformed, missing values were filled by imputation with random numbers drawn from a normal distribution calculated for each sample[132]. Three samples (1 outlier/group in "input", "mock", "dsRNA" and "silvesterol" groups) were excluded from further analysis as they did not pass quality control (PCA analysis). Significantly regulated proteins were

determined by ANOVA with permutation-based FDR statistics (S0 = 0, 250 permutations, FDR ≤ 0.05). Unsupervised hierarchical clustering on z-scored intensities was used to identify significantly regulated proteins upon each treatment. Functional annotations were extracted from UniProtKB, GO, the Kyoto Encyclopedia of Genes and Genomes and CORUM. Pathway enrichment analysis was performed using a one-sided Fisher's exact test with Benjamini-Hochberg correction for multiple hypothesis (FDR cut-off = 0.02). Complete enrichment analysis is available in Supplementary Data 3 and representative categories are visualised. Results were plotted as bar plot and heat map using GraphPad Prism and Perseus[132], respectively. For comparative analysis of input CE proteome with reference "naïve" U2OS global proteome, median-normalised iBAQ intensities of CEs from this study were compared against median-normalised MaxLFQ intensities extracted from a deep U2OS proteome (PRIDE identifier PXD045003). Hypergeometric test was performed for intersections with literature lists, enrichment factor and $p$-values adjusted for multiple testing using the Benjamini-Hochberg procedure to control the FDR are indicated in the corresponding figures. Identification of dsRNA and silvestrol-specific hits was performed using differential abundance analyses comparing dsRNA- or silvestrol-induced condensates with PEG-induced condensates that reflected non-specific enrichment (log2 FC > 2.5). Values are available in Supplementary Data 5.

### Cryo scanning electron microscopy (cryo-SEM)

Cryo-SEM sample preparation was performed as described previously[134]. In brief, 3 µl of the CEOD emulsion solution was dropped onto 0.8 mm-diameter gold specimen carrier assembled on a freeze fracture holder (Leica Microsystems) and immediately immersed in liquid nitrogen. Next, the CEODs were transferred using an evacuated liquid nitrogen-cooled shuttle (Leica EM VCT100, Leica Microsystems) into a freeze fracture and etching system (Leica EM BAF060, Leica Microsystems). CEODs were fractured in a $10^6$–$10^7$ mbar vacuum chamber at −160 °C with a cooled knife. To allow for the sublimation of water in the fractured CEODs, the sample holder stage was heated to −90 °C for 60 min. Following sublimation, the freeze-fractured CEODs were coated with a 4-nm condensed layer of platinum (Pt–C) by electron beam evaporation. For image acquisition, the samples were transferred via an evacuated liquid nitrogen cooled shuttle into the imaging chamber of a Field Emission Scanning Electron Microscope (Zeiss Ultra 55 FE-SEM, Carl Zeiss Microscopy) equipped with in-lens, secondary electron (SE) and angle selective backscattered electron (ASB) detectors (Carl Zeiss SMT). Top-view imaging was performed under low temperature conditions (top = −115 ± 5 °C) and with a working distance between 3 and 5 mm. Due to the low conductivity of the CEOD emulsion low acceleration voltages of 1.5–2.0 kV were used. Signals were detected with the in-lens detector. Uncropped and unprocessed scans are provided in the Supplementary Information file.

### Absolute quantification of G3BP1 protein levels

20 µg of U2OS YFP-G3BP1 CEs were spiked with increasing amounts of recombinant GFP. Samples were subjected to SDS-PAGE (10% polyacrylamide gels), Western blotting, and immunostaining with anti-GFP (1:1000 BSA; Roche) as described above. Band intensities were quantified as described above, and recombinant protein titration intensities were used as standard curves to quantify the mass percentage of YFP-G3BP1 fusion protein in U2OS translation-competent CEs, which reflect a concentration of 0.5 µg/µl YFP-G3BP1 (27 nM).

### Inhibition of YFP-G3BP1 condensation in CEODs by USP10-derived peptides

The biotinylated peptides USP10$_{8-25}$ WT (YIFGDFSPDEFNQFFVTP-biotin) and mutant F10A-F13A YIAGDASPDEFNQFFVTP-biotin) were

synthesised by GenScript (New Jersey, US). 1 μM or 10 μM of biotinylated peptides were added to the translation reaction mix containing 250 mM dsRNA before incubation at 37 °C.

### Loss of function siRNA screen

The expression of 6 validated SG candidates was silenced by reverse transfection using siRNA pools (SmartPool, Dharmacon). 25 nM siRNA (SmartPool, Dharmacon) were mixed with 1 μl RNAi Max (Thermo Fisher Scientific) in a final volume of 100 μl OPTI-MEM medium (Life Technologies) and incubated for 20 min at room temperature. The transfection mix was added to $4 \times 10^4$ U2OS cells resuspended in 500 μl culture medium without antibiotics seeded in a 24-well plate (reverse transfection). Medium was replenished after 24 h. SG disassembly experiments were performed 48 h after silencing.

### SG disassembly

Silenced U2OS cells ($4 \times 10^4$ per well on glass coverslips in 24-well plates) or U2OS FBXO3$_{KO}$ cell pools ($4 \times 10^4$ per well on glass coverslips in 24-well plates) were treated with 0.5 mM arsenite for 45 min or 2 μg/ml silvestrol for 1 h. Medium was removed and cells washed twice with PBS 1× before addition of fresh culture medium. Cells were fixed at the indicated time points after stress removal with 4% PFA in PBS for 15 min and further processed for immunofluorescence analysis as described above.

### Statistics and reproducibility

Statistical analysis was performed using GraphPad Prism software, unless stated otherwise. All testing was performed using two-sided, unpaired testing and assuming normal distribution of data. For quantification of CEODs and condensate features, Kruskal–Wallis test was performed, followed by Dunn's multiple comparison test. Translation and protein expression levels were tested as described below: For comparing two datasets, Student's $t$-test was applied. For comparing three or more datasets with one or more variables, one-way or two-way ANOVA was performed, respectively. For multiple comparisons, the critical α-level was adjusted for 2-way ANOVA with Sidak's multiple comparisons test and for one-way ANOVA with Dunnett's multiple comparisons test (comparison to one ctrl) or Tukey's multiple comparisons test (comparison to all other means). $P$ values $\geq 0.05$ were considered as statistically non-significant (ns).

### Reporting summary

Further information on research design is available in the Nature Portfolio Reporting Summary linked to this article.

## Data availability

The main data supporting the findings described in this manuscript are available in the article, in its Supplementary Information, and Source Data File. The analysis pipeline (segmentation of CEODs and condensates) is available on request to Alessia Ruggieri for import into Nis-Elements software. The mass spectrometry-based proteomics data have been deposited at the ProteomeXchange Consortium (https://proteomecentral.proteomexchange.org/cgi/GetDataset?ID = PXD057906) via the PRIDE partner repository with the following dataset identifier: PXD057906. The RNA sequencing data have been deposited at the NCBI GEO and are accessible with the following identifier: GSE308184. Source data are provided with this paper.

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

## Acknowledgements

We thank all members of our research group for valuable discussions. We thank Georg Stoecklin and Katharina Haneke (Division of Biochemistry, MI3, Heidelberg University, Medical Faculty Mannheim), Claire Riggs and Nancy Kedersha (Brigham and Women's Hospital, Harvard Medical School, Boston) for enlightening and inspiring conversations. We are grateful to Ulrike Herian (Molecular Virology, Heidelberg University) for her support with the maintenance and amplification of cell lines, to Monika Langlotz at the Flow Cytometry & FACS Core Facility (ZMBH Heidelberg University) for her support with cell sorting, to Severina Klaus, Jessica Kehrer, Sylvia Olberg and Vibor Laketa at the Infectious Diseases Imaging Platform (IDIP) (CIID; Heidelberg University) for expert assistance. U.E. acknowledges the financial support of the CellNetworks Core Technology Platform (CCTP) for the Nikon Imaging Center. The Max Planck Society is appreciated for its general support. This work was supported by the Deutsche Forschungsgemeinschaft (DFG, German Research Foundation) project 240245660 (SFB1129 TP13 to A.R., TP15 to J.P.S., and TP8 to O.T.F.) and project 533587280 - Cluster of Excellence "SynthImmune" (EXC3018/1 to A.R., K.J., O.T.F., I.P. and J.P.S.). P.S. was supported by the DFG project 528559282 and by the Free and Hanseatic City of Hamburg and the Federal Ministry of Education and Research (BMBF, VirMScan, project 13GW0622). P.I. was supported by funds of the National Institutes of Health grant R01 GM126150 and R01 GM146997. We acknowledge Heidelberg University for the financial support of publication fees.

## Author contributions

A.R. conceived, designed and supervised the study. A.L., Z.S. and P.K. performed the experiments, analyzed the data, and interpreted the results. K.K. provided expertise in the production of CEs and cloning. O.S. and K.J. performed parts of the microfluidics experiments under the supervision of I.P. and J.P.S. S.H. and P.I. provided expertise on SGs and phase separation. U.E. aided with confocal microscopy and conceived the CEOD imaging and analysis pipeline. N.M. analysed the RNA sequencing data analysis. I.P. performed cryo-SEM measurements. C.F. performed proteomics experiments and data analysis under the supervision of P.S. A.L. and A.R. wrote the manuscript. O.T.F., P.I., P.S. and J.P.S. and A.R. secured funding. All authors reviewed, edited and approved the manuscript.

## Funding

## Competing interests

The authors declare no competing interests.
