## [Transparent Peer Review file · Nature Communications]

Stress granule phase separation in stress-responsive cytosolic extract-in-oil droplets

Corresponding Author: Dr Alessia Ruggieri

Version 0:

Reviewer comments:

Reviewer #1

(Remarks to the Author)

The authors of this publication have developed an in vitro system to measure condensate formation. Cytoplasmic extracts are encased in oil droplets allowing for high protein concentration within the cytoplasmic extracts. Condensation in these extracts can be visualized after initiation of the integrated stress response and inhibition eIF4A. Similarly to previous in vitro and in vivo models, the authors find that both G3BP proteins and RNA is critical for condensate formation. They use mass spectrometry of purified condensates to identify novel stress granule components.

This publication is well written and provides a valuable tool representing innovation concerning the use of in vitro systems to study condensate biology. Overall, the experiments are well designed, and the conclusions are well interpreted. I would recommend publication with a few minor additions / edits to the publication:

Major Concerns

- The primary concern I have is with insufficient detail on the production of the CEODs. Their reproducible generation is a major advancement in the field of in vitro condensate biology and a more detailed protocol for how these oil droplets are generated should be included in either the method section and/or the main body of the publication including catalogue numbers of oils, surfactants, and other reagents used to generate the CEODs. Additionally, specific details about how these droplets were visualized (how was the home-made visualization chamber made) and specific details on how the CEODs were destabilized should be included.
- The condensates formed are partially mobile, but they appear more solid-looking than typical stress granule condensates. Would it be possible to examine the protein solubility biochemically (RIPA vs. Urea) for a constituent protein such as G3BP1 in the presence and absence of the induced stressor?
- The mass spectrometry and the analysis of the data is well done but I have some concern that the MS approach taken may not be entirely representative of condensate composition. The method section indicates that pellets were washed twice following centrifugation. Since there is a high probability that there is liquid fraction within the condensates, it is likely that these washes would eliminate any purely liquid fraction of the condensate, leaving only the solid or gel like fraction. Conversely, centrifugation could lead to solidification of the condensates bringing false positives into the pellet. This could explain why there was high levels of G3BP1/Caprin in the Mock treated CEODs. While these limitations may be unavoidable, the limitations of LC-MS/MS for condensate composition should be noted.

Minor Concerns

Figure 1D:

- Overall, this publication uses a large number of violin plots to represent graphical data. It seems that in many instances such as in figure 1D, a better visually appealing representation of the data would be to have two independent line graphs of the two biological replicates with the average intensity +/- either standard deviation or error. Additionally, the biological differences appear muted in some graphs such as Supp Fig. 15C where the outlier data account for much of the visual space on the graph.

Supp. Figure 1A:

- Figure legend should state top/bottom panel rather than left/right

Figure 2C:

- Were the CEs or CEODs treated with dsRNA, ssRNA or DNA? There appears to be a discrepancy between the text body and figure legend.

Line 175:

- There is a callout to figure 2C that appears to be for figure 2D instead

Figure 3/5:

- It would be helpful to see the formation of condensates in a cellular context side by side with the CEODs following addition of the indicated stressor.

Supp. Figure 4A:

- The number of condensates analyzed per biological replicate is missing from this figure (or legend)

Figure 6:

- The concentration of nuclease used should be specified in the figure legend

Reviewer #2

(Remarks to the Author)

Summary:

Biomolecular condensates are widely found in intracellular, extracellular and two-dimensional membranes, providing subcellular compartmentalization. Their involvement in numerous biological processes necessitates the development of a scalable system resembling the cellular complexity. In this regard, the tool described in this manuscript by Reuter et al., is a compelling research tool. The authors showcase their design of cytosolic extracts in oil droplets (CEODs) to study stress granule phase separation.

Using insults known to induce stress granules, like dsRNA and eIF4 poisons, Reuter et al., show that the CEODs respond to stress and induce stress granule formation via intact signaling pathways and scaffolds (G3BP1 and G3BP2). To demonstrate the utility of CEODs, the authors show that stress granules formed in CEODs respond to inhibitors (e.g., USP10 peptide that binds to G3BP1 NFT2 domain) and characterize the proteomes of stress granules formed via distinct inducers. While this tool is promising, the lack of a thorough characterization of the CEODs derived from the mechanical disruption of cells in hypotonic stress and the lack of specificity in stress granules formed in CEODs undermine the reliability of this system as a substitute for in-cell studies.

Major:

1. This manuscript can be strengthened by characterizing CEODs in a systematic fashion to support their claim that CEODs are 'confined, cell-free system that retains responsiveness under near-to-normal physiological conditions'. Can authors provide a thorough characterization of the CEODs? Describe the composition and abundance of RNAs and proteins in CEODs and characterize the extent of the crowding environment created by the oil droplet system in CEODs. This is important given that condensate formation is regulated by the abundance of RNAs, proteins and the level of molecular crowding.

First, how do the protein concentration and transcript concentration compare to the intracellular environment? Can authors provide a distribution of transcript abundance and protein abundance ranges in CEODs?

Second, how do the transcriptome and proteome from CEODs generated from cellular lysates collected from hypotonic treatment differ from existing references of the transcriptome and proteome? There is concern of altered cell state due to the hypotonic stress. Also can authors compare CEODs to the cell extracts (prior to embedding in to oil droplets) to show that the process of embedding lysate extracts in oil droplets induced marginal changes? Lastly, the oil droplet system implemented in CEODs – to what extent of crowding is induced by this system?

2. Show specificity of condensate proteomics. Authors report 5-6K proteins in stress granules formed in CEODs. These numbers are ~50% of detectable total proteome in a single data-dependent acquisition mode. Either by improving the stress granule extraction method or improving steps in the quantitative analysis of the proteomics data, authors need to show that condensates formed in CEODs display the specificity of stress granule proteome. Data presented (Fig 9) show a wide range of GO terms that are not associated with stress granule proteome (e.g., mitochondrial genome maintenance) and lack clear relevant terms related to stress granules like RNA-binding and translation.

3. After improving the proteomics report of stress granule proteome, please include a comparison to cell lysate stress granule proteome described in Freibaum BD et al., JCB 2021.

4. Ultrastructural changes in CEODs may not be physiologically relevant given that the activity of regulators of condensates (e.g., chaperones and autophagy) are not measured in this study (chaperones, ATPs etc). Please tone down the interpretation in line 289, 'reminiscent of the formation of nanofibrils in droplets containing protein microgels.' Due to the lack of specificity in CEOD stress granules proteome, I am not sure that the ultrastructure changes observed in different conditions can be interpreted in a meaningful way.

5. Can authors discuss reasons for the differences observed in CEODs generated from different cell types? For example, Fig S4a: dsRNA induced condensates' penetrance varies between U2OS and Huh7 cells. Will this be important for readers to consider the appropriate cell type for CEODs?

Minor:

1. Line 60 in the Introduction section: rapamycin does not induce stress granules. The reference in this sentence does not report this. Instead, the reference reports the sequestration of Astrin, a regulator of mTOR in stress granules.

2. Please explain why authors chose 15 μm diameter CEODs, while the average is 25.

3. Line 194: Authors say that stress granules are less circular in CEODs compared to those formed in cells. Can authors provide the circularity in cells and provide a reference?
4. Related to USP10 peptide inhibition, how does the USP10 peptide to G3BP1 compare to the endogenous molar ratio between G3BP1 and USP10 in U2OS cells?
5. Supplementary Fig9a: Nuclease treatment does seem to have an effect on translation efficiency in Huh7 cells. However, this is noted as non-significant. Please explain.
6. Stress granule induction assays often involve oxidative stress. Can authors note if oxidative stress works to induce stress granules in CEODs?

Version 1:

Reviewer comments:

Reviewer #1

(Remarks to the Author)

This revision has predominantly satisfied my previous concerns.

In the following sentence (line 684-686): "Fifteen μ l of translation reaction mix were layered on a mixture of fluorinated oil (Novec 7500, >99%, 3M) and a PEG-based fluorosurfactant (008-FluoroSurfactant-1G, Ran Biotechnologies)", the ratio of fluorinated oil to PEG-based fluorosurfactant in the mixture should be stated.

With regards to the identification of proteins by mass spectrometry, I do share reviewer 2's concerns. While this approach favors the inclusion of "shell" proteins identified in their purified mass spectrometry sample, it does so at the cost of increased false positives (increased SG proteome) found in the Mock samples. The authors do address these issues in the discussion section, but perhaps greater attention could be given to this issue. As the authors' state in their rebuttal letter, there is no unbiased method for the purification of SG, so any method has drawbacks. Regarding the inclusion of mitochondrial proteins in the dsRNA sample, this could be due to inclusion of these proteins in the condensate or a false positive due to the inclusion of pelleted mitochondria at high speed. Would it be possible to visualize whether these proteins localize within the CEOD condensate following treatment with dsRNA?

Reviewer #2

(Remarks to the Author)

Major

1. As it stands, claiming that over half of the proteome (6,060 proteins) is associated with stress granules diminishes the excitement and value of the CEOD system. The lack of specificity is likely due to the experimental method used—simple centrifugation of cell extracts (CEs)—to isolate condensate-associated proteins. I encourage the authors to utilize the quantitative information identified across 4-5 replicates in each condition to determine a very stringent threshold that shows significant changes. Please provide a list of candidates that can be utilized as a resource. The authors can use the same criteria used to narrow down potential candidates that were followed up on. For example, authors can provide a high-confidence list based on the fold-change observed between mock and treatments that induce physiologically relevant stress granules in CEODs (e.g., Silversterol and dsRNA), but leave out PEG, which shows completely different enrichment of transcriptome. Once more stringent proteomics data are presented, I recommend repeating the GO enrichment analysis and comparison to published proteomes.
2. Justification of the quality of the proteomics data is poorly done. In Fig. 9C, the authors used this large data set (6,060 proteins) to measure recovery of the known standard stress granule proteome (Tier 1 in RNA granule), stress granule core, and lysate granules. This is an inappropriate comparison, since the size of the starting dataset is so large. The authors should test if this overlap is higher than random using methods like the hypergeometric test. I also recommend the use of the updated stress granule proteome (407 proteins) reported in Millar et al., 2023.
3. In Figure 8d, the authors have used the full transcriptome (14,324 transcripts) identified instead of the stress granule-enriched transcriptome (1,626) from Khong et al., 2017. This comparison to the full reported transcriptome is inappropriate. Please compare to the stress granule-enriched transcriptome as reported in Khong et al., not the entire transcriptome. Similar to my previous comment, given the large number of transcripts identified, please use statistical methods like the hypergeometric test to show significance.
4. The authors report that the CEOD transcriptome and proteome of the mock condition were already enriched for known components, suggesting pre-existing stress granule seeds. An alternative explanation is that the mock CEODs activated the integrated stress response during CE isolation using hypotonic stress. In the rebuttal figure 2, mock CE and CEODs show an elevated p-eIF2a level, indicating moderate activation of the integrated stress response. This figure also shows activation of p-p38 in mock-treated CE CEODs. Given that the main purpose of this manuscript is to report a new CEOD system for studying stress granules, these data should be included in the manuscript as part of CEOD characterization.

Minor: Supplementary Figure 13b shows a comparison of the total proteome vs. the CEOD proteome. While most of the indicated stress granule proteins (e.g., G3BP1, CAPRIN and novel stress granule proteins) are similar, USP10 is significantly less in CEOD. This is worth mentioning in the main manuscript, since USP10 is a negative regulator of stress granule formation (Kedersha et al., JBC 2016, and Sanders et al., Cell 2020).

Version 2:

Reviewer comments:

Reviewer #1

(Remarks to the Author)

The authors have addressed all of my concerns through additional experiments and/or text edits. The data showing MGME1 in cellular SGs is especially strong.

Reviewer #2

(Remarks to the Author)

The authors have added new data and analysis of the transcriptome and proteome data in the revised manuscript, which did improve on some areas. However, the authors have not appropriately addressed my concern about presenting 6K proteins as 'proteins associated with the condensates'.

In the revised manuscript, the authors still interpret the 6K proteins identified across all conditions as candidates for the stress granule proteome, as stated in the results section: "The quantitative proteomic analysis of condensates identified a remarkably high number of proteins associated with the condensates (Supplementary Fig. 14a, Supplementary Table 5)."

Currently, the human proteome detectable using highly sensitive mass spectrometers ranges between 7~8K (timsTOF pro) to 10K (Thermo Astral). Interpreting 60% of the detectable proteome as the potential condensate proteome is not scientifically sound.

To provide perspective, the 5~6K proteins that the authors propose as potential condensate proteome are 10~20x higher than those reported by other published methods. Furthermore, the lack of selectivity and specificity is evident in their enrichment analysis; dsRNA and silvestrol-induced proteome shows negligible enrichment factors, ranging from 1.08 to 1.14 (Supplementary Figure 14b).

The authors argue that the quantitative analysis provided in Supplementary Table 5 should be mined as a resource for the community. I reviewed the table again. Limiting the proteins that are significantly different between physiologically relevant stressors (dsRNA or silverstrol) and the crowding agent, PEG—using filters provided for the Column 25 (ANOVA significant) and Column 35 (Student's T-test significant)—still revealed a large number, 5,086 proteins. This number even exceeds the estimated number of RNA-binding proteins in the human proteome (~4K).

I disagree with the authors' claim that the dataset provides a useful resource for the condensate community.

Reviewer #1 (Remarks to the Author):

The authors of this publication have developed an in vitro system to measure condensate formation. Cytoplasmic extracts are encased in oil droplets allowing for high protein concentration within the cytoplasmic extracts. Condensation in these extracts can be visualized after initiation of the integrated stress response and inhibition eIF4A. Similarly, to previous in vitro and in vivo models, the authors find that both G3BP proteins and RNA is critical for condensate formation. They use mass spectrometry of purified condensates to identify novel stress granule components.

This publication is well written and provides a valuable tool representing innovation concerning the use of in vitro systems to study condensate biology. Overall, the experiments are well designed, and the conclusions are well interpreted. I would recommend publication with a few minor additions / edits to the publication:

We thank the reviewer for the very positive evaluation of our manuscript.

Major Concerns

- The primary concern I have is with insufficient detail on the production of the CEODs. Their reproducible generation is a major advancement in the field of in vitro condensate biology and a more detailed protocol for how these oil droplets are generated should be included in either the method section and/or the main body of the publication including catalogue numbers of oils, surfactants, and other reagents used to generate the CEODs. Additionally, specific details about how these droplets were visualized (how was the home-made visualization chamber made) and specific details on how the CEODs were destabilized should be included.

We apologize for the previous lack of clarity and have added detailed information about oil and surfactant and all other reagents in the Methods section to improve reproducibility. We also included a clearer description of the homemade visualization chamber and the protocol used to destabilize the CEODs (page 15, lines 681–691).

- The condensates formed are partially mobile, but they appear more solid-looking than typical stress granule condensates. Would it be possible to examine the protein solubility biochemically (RIPA vs. Urea) for a constituent protein such as G3BP1 in the presence and absence of the induced stressor?

We thank the reviewer for this suggestion. To address this, we now evaluated the solubility of SG core proteins G3BP1, TIA-1, and Caprin1 in dsRNA- and silvestrol-induced condensates by pelleting the condensates, resuspending them in RIPA buffer, centrifuging, and then resuspending the RIPA-insoluble fraction in urea buffer. No change in the solubility of these proteins was observed in CE compared to the RIPA- and urea-soluble fractions. This finding aligns with the previous report by Kedersha and colleagues, who used b-isox precipitation to assess protein solubility in arsenite-induced SGs and found no change in the solubility of SG proteins TIA-1, TIAR, and G3BP1 compared to control cells (Kedersha et al. 2016, PMID: 27022092). This result is now included in the main text (page 6, lines 274-275) and shown in Supplementary Fig. 11a.

- The mass spectrometry and the analysis of the data is well done but I have some concern that the MS approach taken may not be entirely representative of condensate composition. The method section indicates that pellets were washed twice following centrifugation. Since there is a high probability that there is liquid fraction within the condensates, it is likely that these washes would eliminate any purely liquid fraction of the condensate, leaving only the solid or gel like fraction. Conversely, centrifugation could lead to solidification of the condensates bringing false positives into the pellet. This could explain why there was high levels of G3BP1/Caprin in the Mock treated CEODs. While these limitations may be unavoidable, the limitations of LC-MS/MS for condensate composition should be noted.

We thank the reviewer for these valuable thoughts. The limitations primarily lie in the methods used to assess SG composition rather than in the LC-MS/MS technique itself. In fact, no unbiased method exists for purifying SGs. Different isolation and analysis approaches can influence the detected components.

Proximity labelling or pull-down experiments targeting the SG core protein G3BP1 primarily capture its interaction network, whereas the pelleting approach with two washing steps and high-speed centrifugation we designed was intended to also include shell proteins. Although this method has limitations, as noted by the reviewer, the strong overlap between our condensate datasets and high-confidence SG components listed in the RNA Granule Database, which integrates proteins identified by multiple approaches, gives us confidence in its value. This method has enabled us to identify six novel SG components and a further functional screen has revealed FBXO3 as a regulator of SG disassembly dynamics (Fig. 10b). This aspect is now addressed in the Discussion section (pages 9-10, lines 413-417). Although the number of proteins detected in pelleted mock samples was large, this finding was not unexpected. Several studies have demonstrated the existence of pre-formed SG core protein networks, often referred to as SG seeds, even in the absence of stress, supporting the model that these seeds assemble into larger condensates upon stress induction (Markmiller et al., 2018 PMID: 29373831; Youn et al., 2018 PMID: 29395067; Yang et al., 2020 PMID: 32302571). The fact that we detected all SG core components without significant enrichment in dsRNA- or silvestrol-induced condensates strongly reinforces this concept. Concerns addressing the LC-MS/MS technique and analysis are addressed in more detail in response to Reviewer 2 point 2.

Minor Concerns

Figure 1D:

- Overall, this publication uses a large number of violin plots to represent graphical data. It seems that in many instances such as in figure 1D, a better visually appealing representation of the data would be to have two independent line graphs of the two biological replicates with the average intensity +/- either standard deviation or error. Additionally, the biological differences appear muted in some graphs such as Supp Fig. 15C where the outlier data account for much of the visual space on the graph.

We selected violin plots because they effectively represent the underlying data distribution, including density profiles and potential “outliers”. We would also like to emphasize that “outliers”, such as those observed in the previous Supplementary Fig. 15c (now Supplementary Fig. 19c), are an integral part of the data as they reflect meaningful biological variability within the system. However, we acknowledge the reviewer's point that this type of visualization does not adequately convey differences between biological replicates. To address this, we now included additional line graphs displaying the mean and standard deviation corresponding to each violin plot of condensate area across the main CEOD types: new Fig. 3d for dsRNA-induced condensates, Fig. 5c for silvestrol-induced condensates, and Supplementary Fig. 5b for PEG-induced condensates. This complementary visualization better highlights the inherent variability and demonstrates the comparability of biological replicates.

In addition, the Rebuttal Fig. 1 below presents a line graph representation of the eGFP translation data shown in Fig. 1e and the previously reported data in Supplementary Fig. 15c (now Supplementary Fig. 19c).

Rebuttal Fig. 1. Experimental reproducibility. The condensate area measurements were reanalyzed to visualize the variability in the experimental results between two biologically independent replicates. The data are presented as line graphs showing the mean \pm SD for each replicate. The number (n) of CEODs analyzed per biological replicates is indicated at the top. **(a)** Translation of an eGFP reporter transcript in CEODs over 8 h at 37 °C (corresponds to Fig. 1e). The line graph represents the eGFP signal intensity in CEODs (eGFP IVT). CEODs generated without eGFP reporter RNA (no IVT) were used as control of background autofluorescence. **(b)** Mean condensate area displayed for two individual replicates for the validation of new cell lines expressing the corrected YFP-G3BP1 (previous Supplementary Fig.15c, now displayed as Supplementary Fig. 19c).

Supp. Figure 1A:

- Figure legend should state top/bottom panel rather than left/right

This was changed accordingly.

Figure 2C:

- Were the CE or CEODs treated with dsRNA, ssRNA or DNA? There appears to be a discrepancy between the text body and figure legend.

We apologize for the previous lack of clarity. CE were treated with either dsRNA, ssRNA, or DNA. We have rephrased the sentence in the main text: "To assess the stress responsiveness and SG-associated LLPS in CEODs, U2OS and Huh7 CE were treated with double-stranded (ds) RNA, ..." (page 4, lines 166-167).

Line 175:

- There is a callout to figure 2C that appears to be for figure 2D instead

This was changed accordingly.

Figure 3/5:

- It would be helpful to see the formation of condensates in a cellular context side by side with the CEODs following addition of the indicated stressor.

We thank the reviewer for this valuable suggestion. In response, we have added an image panel displaying condensates in CEODs and SGs in cells induced by the same triggers. Fig. 3e shows condensates and SGs induced by dsRNA. Fig. 5d shows condensates and SGs induced by silvestrol. Additionally, as also requested by Reviewer 2, we conducted a detailed characterization and comparison of condensate size, including perimeter and circularity. These data are now presented in Fig. 3e, 3f, Fig. 5d, and new Supplementary Table 2 ("Comparison-condensates-vs-SGs"). Furthermore, information on volume, protein concentration, and RNA concentration is provided in new Supplementary Table 1 ("Comparison-cells-vs-CEODs").

Supp. Figure 4A:

- The number of condensates analysed per biological replicate is missing from this figure (or legend)

We appreciate the reviewer's attention to this detail. The previous Fig. 4a represented the percentage of CEODs with condensates (2 biological replicates) in PKR knockout CEODs. The number of CEODs analysed was displayed in the main Fig. 4b. Since the original presentation of the data was unclear, we have now restructured Fig. 4, which focuses on dsRNA, to display both panels side by side for better clarity. Data on PEG-induced condensates, used as control for translation repression unrelated condensates, are now presented in the new Supplementary Fig. 4.

Figure 6:

- The concentration of nuclease used should be specified in the figure legend

The concentration of micrococcal nuclease used (18 U) is now also specified in the figure legend (Fig. 6c-f).

Reviewer #2 (Remarks to the Author):

Summary:

Biomolecular condensates are widely found in intracellular, extracellular and two-dimensional membranes, providing subcellular compartmentalization. Their involvement in numerous biological processes necessitates the development of a scalable system resembling the cellular complexity. In this regard, the tool described in this manuscript by Reuter et al., is a compelling research tool. The authors showcase their design of cytosolic extracts in oil droplets (CEODs) to study stress granule phase separation.

Using insults known to induce stress granules, like dsRNA and eIF4 poisons, Reuter et al., show that the CEODs respond to stress and induce stress granule formation via intact signaling pathways and scaffolds (G3BP1 and G3BP2). To demonstrate the utility of CEODs, the authors show that stress granules formed in CEODs respond to inhibitors (e.g., USP10 peptide that binds to G3BP1 NFT2 domain) and characterize the proteomes of stress granules formed via distinct inducers.

While this tool is promising, the lack of a thorough characterization of the CEODs derived from the mechanical disruption of cells in hypotonic stress and the lack of specificity in stress granules formed in CEODs undermine the reliability of this system as a substitute for in-cell studies.

Major:

1. This manuscript can be strengthened by characterizing CEODs in a systematic fashion to support their claim that CEODs are 'confined, cell-free system that retains responsiveness under near-to-normal physiological conditions'. Can authors provide a thorough characterization of the CEODs?

Describe the composition and abundance of RNAs and proteins in CEODs and characterize the extent of the crowding environment created by the oil droplet system in CEODs. This is important given that condensate formation is regulated by the abundance of RNAs, proteins and the level of molecular crowding.

We agree with this reviewer on the importance of crowding and the importance of creating an environment that come as close as possible to the cytosol. We made a thorough and careful characterization of the condensates in CEODs and compared their volume, protein, and RNA concentration to SGs in cells induced by the same triggers. This information is now included in Supplementary Fig. 5a, b, new Supplementary Table 1 ("Comparison-cells-vs-CEODs") and Supplementary Table 2 ("Comparison-condensates-vs-SGs").

First, how do the protein concentration and transcript concentration compare to the intracellular environment? Can authors provide a distribution of transcript abundance and protein abundance ranges in CEODs?

The average concentrations of protein and RNA per cell were estimated from a defined number of cells prior to lysis and extraction in order to measure total concentrations and relate them to a single cell.

Protein and RNA concentrations in CEODs was extrapolated based on the CE protein concentration used and their diameter. Results for triplicate measurements (mean \pm SD) are now included in new Supplementary Table 1 (“Comparison-cells-vs-CEODs”). Importantly, Supplementary Fig. 1d shows a uniform distribution of YFP-G3BP1 in CEODs, as shown by a positive correlation between YFP-G3BP1 mean fluorescence intensity and CEOD diameter (Supplementary Fig. 1f).

Second, how do the transcriptome and proteome from CEODs generated from cellular lysates collected from hypotonic treatment differ from existing references of the transcriptome and proteome?

To address this comment and to compare protein concentrations in U2OS CEODs with those in the intracellular environment, we compared the median-normalized read count distribution of the proteins identified in our input CE sample or in the condensates with those of a U2OS global (whole cell) proteome made publicly available on PRIDE (PRIDE identifier PXD045003). Unfortunately, other studies on SG proteomes did not make their input raw data available for direct comparison. These results are now included in new Supplementary Fig. 13. We found no major differences in overall protein abundance profiles, indicating that although only the cytosolic part was analysed, our candidate selection was not skewed toward highly abundant or rare proteins. Importantly, the correlation plot comparing the published dataset with our input CE measurement (Supplementary Fig. 13b) showed that both known and newly identified SG proteins cluster near the diagonal, indicating that their enrichment within condensates results from specific recruitment rather than differences in baseline abundance. Taken together, these analyses suggest that the proteome of CEODs represent a physiologically relevant subset of the intracellular environment rather than an artifact of extreme abundance.

To compare the condensate transcriptome with previously published SG transcriptome, we performed transcriptome profiling of the condensates. Three biological replicates of condensates induced by dsRNA, silvestrol, PEG, or untreated controls (mock) were pelleted following the same procedure used for LC-MS/MS. The resulting pellets were resuspended in TRIzol for total RNA extraction. mRNAs were then enriched via poly(A) selection and subjected to RNA sequencing. Results are now included as a new section of the main text (page 7, lines 295-313) and shown in new Fig. 8c-f, Supplementary Fig. 12, Supplementary Table 3 “Condensates-RNAseq-raw-counts”, and Supplementary Table 4 “RNAseq-differential-expression-analysis”). These new results show the enrichment or depletion of several mRNAs relative to total RNA (Fig. 8c). The comparison analysis with the previously published transcriptome of arsenite-induced SGs by Khong et al., 2017 (PMID: 29129640) revealed a striking 94% overlap (Fig. 8d, Supplementary Fig. 12a). Consistently, selected mRNAs were enriched or depleted in the condensates based on their length (Fig 8e, f). Noteworthy, while the transcriptome of PEG-induced condensates exhibited an entirely opposite profile, the transcriptome of mock condensates showed no significant differences with those of dsRNA- and silvestrol-induced condensates. This suggests that a subset of mRNAs enriched in condensates likely pre-associates with SG seeds under normal conditions, i.e. prior to translation repression and LLPS (Supplementary Fig. 12b).

We thank the reviewer for raising this important point, the additional data generated to address this issue significantly strengthened our manuscript. Our analyses demonstrate that the condensate transcriptome induced by translation repression in the CEOD system accurately reflects the RNA composition of SGs assembled in cells and is distinctly different from that of condensates induced by a molecular crowder.

There is concern of altered cell state due to the hypotonic stress. Also can authors compare CEODs to the cell extracts (prior to embedding in to oil droplets) to show that the process of embedding lysate extracts in oil droplets induced marginal changes?

The protocol we used to generate translation-competent cellular extracts is a well-established method widely adopted in the translation field, originally developed by the Hentze laboratory (Rakotondrafara and Hentze, 2011, PMID: 21527914). As shown in Rebuttal Fig. 2, we evaluated the degree of hypotonic stress by measuring the levels of the phosphorylated forms of Akt, p38, and the downstream target HSP27, all of which are indicative of cellular signaling activation in response to hypotonic stress. To rule

out the possibility that embedding of U2OS CEs into CEODs induced hypotonic stress, we compared CEs before and two hours post encapsulation in CEODs. Results were also compared to U2OS whole extracts obtained by scraping cells in ice-cold protein lysis buffer (as detailed in the Methods section “Western blot analysis”). No differences were observed in the levels of phosphorylated Akt, p38, and HSP27 between CEs and CEODs under both mock and dsRNA-treated conditions. Similarly, levels of phosphorylated eIF2 α and phosphorylated PKR increased in response to dsRNA compared to mock but remained unchanged between CEs and CEODs. These results indicate that the encapsulation of CEs into CEODs does not induce hypotonic stress. As expected, because our method for producing translation-competent CEs includes a step of hypotonic swelling, both CEs and CEODs exhibited elevated levels of phosphorylated p38 and phosphorylated eIF2 α compared to whole cell lysates. Importantly, this increase did not compromise CE translation competence or the ability to repress translation in response to dsRNA in CEs.

Rebuttal Fig. 2. Translation remains unaffected by encapsulation in CEODs. Analysis of PKR, eIF2 α , HSP27 and p38 basal and phosphorylated (p-PKR, p-eIF2 α , p-HSP27 and p-p38) levels in U2OS cells, CEs and CEODs. (a) representative Western blot (n=3). (b) Corresponding quantifications. Levels of p-PKR, p-eIF2 α , p-HSP27 and p-p38 were normalized to the basal form and are expressed as arbitrary units.

Lastly, the oil droplet system implemented in CEODs – to what extent of crowding is induced by this system?

The total concentration of proteins, nucleic acids, and other macromolecules in cells was estimated to be of approximately 50-400 mg/ml. None of the individual macromolecular species is present in high concentration, but all species together occupy a significant proportion of the cytoplasm and are therefore referred to as “crowded” (Minton, 1983, PMID: 6633513; Minton, 2001, PMID: 11279227). The protein concentration in CEODs (5 $\mu\text{g}/\mu\text{l}$) is over 20 times more diluted than what we estimated in cells (128 \pm 20 $\mu\text{g}/\mu\text{l}$), whereas the confinement size is similar to the size of cells (new Supplementary Table 1 “Comparison-cells-vs-CEODs”). This suggests that the confinement induced by the CEODs will not create an as crowded environment as observed in cells. In addition, no self-condensation of YFP-G3BP1 was detected microscopically in mock CEODs. Condensation occurred only after treatment with dsRNA or silvestrol followed by incubation at 37 $^{\circ}\text{C}$ for at least one hour, or upon the addition of PEG, which induces molecular crowding through volume exclusion within CEODs, even at 4 $^{\circ}\text{C}$.

However, to rule out that crowding is the main driver of condensate formation on our CEODs, we used fluorescence recovery after photobleaching (FRAP), a method commonly used to determine the kinetics of diffusion of proteins in cells, and therefore possible crowding effects. Measurements of G3BP1 mobility showed similar levels in both mock CEs and CEODs, indicating that encapsulation into CEODs does not cause measurable crowding. This result is now discussed in the main text (page 4, lines 149-150) and shown in Supplementary Fig. 1, new panels g and h.

2. Show specificity of condensate proteomics. Authors report 5-6K proteins in stress granules formed in CEODs. These numbers are ~50% of detectable total proteome in a single data-dependent acquisition

mode. Either by improving the stress granule extraction method or improving steps in the quantitative analysis of the proteomics data, authors need to show that condensates formed in CEODs display the specificity of stress granule proteome. Data presented (Fig 9) show a wide range of GO terms that are not associated with stress granule proteome (e.g., mitochondrial genome maintenance) and lack clear relevant terms related to stress granules like RNA-binding and translation.

We appreciate the reviewer's comment but respectfully disagree regarding the specificity of our proteomic approach. As noted in our response to Reviewer 1, while we cannot completely exclude the loss of some SG components or the potential enrichment of false positives due to the pelleting and washing steps, the comparative analysis of CEODs composition across functionally distinct stressors identified unique specific clusters, allowing us to rule out an overall unspecific or method-dependent condensation of proteins (Fig. 9a). This specificity is particularly evident in the case of PEG-induced condensates that enrich proteins particularly abundant in the input CEs, while dsRNA treatment enriches a unique set of low abundant proteins that are not detectable in the input CE (Supplementary Fig. 14, panels b and c). Notably, among the six SG components we identified as novel SG components, HERC3, MGAT5 and TRIM37 were not detectable in the input CE (Supplementary Fig. 14d). In addition, we identified an almost 80% overlap with Tier 1 SG components from the RNA Granule Database, whereas other methods, such as the lysate granules established by Freibaum et al., 2021 (PMID: 33502444), showed only about a 24% overlap (now included for comparison in Fig. 9c). Finally, this approach combined with unbiased proteomic analysis led to the discovery of six novel SG components, including the E3 ligase FBXO3, which we have now further characterized as a regulator of SG disassembly dynamics. This result is now included in the main text (page 9, lines 372-385) and shown in new Fig. 10c and Supplementary Fig. 17. Altogether, this methodology offers distinct complementary advantages over existing approaches and represents a valuable resource for the research community.

We acknowledge the reviewer's surprise regarding the presence of several GO terms among the top enriched categories that are typically "not associated with the SG proteome". To provide an unbiased representation, we chose to show only the top enriched Gene Ontology Biological Process (GOBP) terms, including mitochondria-related categories, in the main figure. Importantly, GOBP terms directly relevant to SG biology, such as RNA binding and translation-related processes, were also identified and reported in the previous Supplementary Table 2 (renamed Supplementary Table 6). These terms, however, did not display the highest enrichment scores. Therefore, to improve clarity, we have now replaced the bar graphs with dot plots that display all enriched GOBP terms according to their enrichment factor (new Fig. 9b). We marked the three most enriched GOBP terms of each cluster in blue and seven GOBP terms commonly identified in the RNA Granule Database, as defined by Millar et al., 2023 (PMID: 36662637) in red.

We would also like to emphasize that mitochondria-related GOBP terms were predominantly enriched in the dsRNA-induced condensates, for which no established SG proteome currently exists for direct comparison. A link between mitochondria and SGs has been highlighted in other studies and conditions. For example, SGs induced by metabolic stress were reported to modulate mitochondrial functions and regulate mitochondrial permeability to fatty acids. Analysis of their interactome revealed the presence of mitochondria-associated proteins, supporting their role in metabolic adaptation through direct interaction with mitochondria (Amen and Kaganovich D, 2021, PMID: 34133922). In addition, infection with RNA viruses that generate double-stranded RNA during replication frequently induce mitochondrial remodelling (Sorouri et al., 2022, PMID: 35073751) and activate the mitochondrial-associated adaptor protein MAVS, a key component of the antiviral immune response (Eiermann et al., 2020 PMID: 32899736). Thus, while mitochondrial-related GO terms appear in our analysis, they do not undermine the presence of key SG-related terms in the proteomes, but rather highlight the complexity and specificity of the stress response, particularly in response to dsRNA. This is now discussed in the revised manuscript (p10-11, lines 465-483).

3. After improving the proteomics report of stress granule proteome, please include a comparison to cell lysate stress granule proteome described in Freibaum BD et al., JCB 2021.

As requested by the reviewer, we compared the condensate proteomes with that of lysate granules generated by the addition of recombinant G3BP1 to U2OS cell lysates at concentrations that induce self-condensation. Although this method by Freibaum et al. shares conceptual similarities with CEODs, it lacks essential cellular processes such as translation repression and stress signalling activation, both of which are crucial for SG formation *in vivo*. This is also reflected by a significantly lower overlap (24%) of the lysate granule proteome with high-confidence SG proteins listed in the RNA Granule Database Tier 1. Consistently, the proteome of dsRNA-induced condensates which form in response to the ISR activation only exhibit an overlap of approximately 27% compared to lysate granules. This comparative analysis has been now included in Fig. 9c and Supplementary Fig. 14e and is discussed in the revised manuscript (pages 10-11, lines 456-464).

4. Ultrastructural changes in CEODs may not be physiologically relevant given that the activity of regulators of condensates (e.g., chaperones and autophagy) are not measured in this study (chaperones, ATPs etc). Please tone down the interpretation in line 289, 'reminiscent of the formation of nanofibrils in droplets containing protein microgels.' Due to the lack of specificity in CEOD stress granules proteome, I am not sure that the ultrastructure changes observed in different conditions can be interpreted in a meaningful way.

Both transcriptome and proteome of the condensates, as well as the identification and validation of several new SG components, strongly support a genuine SG-like condensation in CEODs in response to dsRNA or silvestrol treatment or upon molecular crowding induced by PEG. Here, we specifically highlight the work by the Knowles laboratory, which produced similar cryo-SEM images of microgels formed from nanofibril-forming proteins using the same water-in-oil encapsulation and droplet-based method. We acknowledge that this observation is purely qualitative and have revised the wording to prevent any over-interpretation (page 7, line 288-293).

5. Can authors discuss reasons for the differences observed in CEODs generated from different cell types? For example, Fig S4a: dsRNA induced condensates' penetrance varies between U2OS and Huh7 cells. Will this be important for readers to consider the appropriate cell type for CEODs?

We thank the reviewer for this important question. The selection of cell lines is a critical step for subsequent functional analyses. The choice of the appropriate cell line should be first guided by the specific tissue or cell type in which SG phase separation is to be studied. Notably, SG composition varies not only with the type of stress applied but also likely depends on the cell type, e.g. cancer cells or neurons.

The difference in penetrance highlighted by the reviewer is a key consideration. In both cells and CEODs, G3BP1 LLPS likely depends on the extent of translational repression, which is influenced by the levels of the targeted stress kinase as well as the abundance of G3BP1 and other core regulatory partners such as Caprin1 and USP10. We included now in Supplementary Fig. 2a the comparative analysis of Huh7 and U2OS cells. Compared to U2OS cells, Huh7 cells express comparable levels of G3BP1 and eIF2 α , but lower levels of PKR and higher levels of some SG core proteins, such as Upf1, Caprin1, and USP10. This latter difference may also influence the threshold for LLPS. However, since Caprin1 and USP10 regulate G3BP1 and SG formation through mutually exclusive interactions with G3BP1, and both are similarly elevated in Huh7 cells, we suppose that the greater sensitivity of Huh7 cells to dsRNA is mainly due to their lower basal PKR expression levels. We previously modelled PKR activation in response to increasing concentrations of dsRNA in Huh7 cells, revealing a bell-shaped activation curve. At higher dsRNA concentrations, PKR kinase activity is inhibited because dsRNA saturates PKR monomers, preventing the formation of active dimers and oligomers. This activation curve shifts when PKR levels are elevated (Klein et al., 2022, PMID: 35319985).

We have incorporated the comparison of the cell lines into the main text (page 4, lines 196-198) and Discussion section (page 10, lines 434-438). We hope these considerations will help readers in selecting the most appropriate cell type for their studies.

Minor:

1. Line 60 in the Introduction section: rapamycin does not induce stress granules. The reference in this sentence does not report this. Instead, the reference reports the sequestration of Astrin, a regulator of mTOR in stress granules.

We thank the reviewer for identifying the incorrect citation. The sentence has been removed to address this issue.

2. Please explain why authors chose 15 μm diameter CEODs, while the average is 25.

We included for analysis CEODs with a diameter ranging from 15 μm to 100 μm in diameter to reflect a median diameter ($25.3 \pm 13.1 \mu\text{m}$) comparable to that of Huh7 and U2OS cells (Supplementary Fig. 1c, 1d, Supplementary Table 1).

3. Line 194: Authors say that stress granules are less circular in CEODs compared to those formed in cells. Can authors provide the circularity in cells and provide a reference?

We thank the reviewer for this valuable suggestion. As mentioned in response to Reviewer 1, we have added a comparison of the perimeter and circularity of condensates and SGs induced by dsRNA or silvestrol. These data are now presented in Fig. 3e, 3f, Fig. 5d, and Supplementary Table 2 (“Comparison-condensates-vs-SGs”).

4. Related to USP10 peptide inhibition, how does the USP10 peptide to G3BP1 compare to the endogenous molar ratio between G3BP1 and USP10 in U2OS cells?

We extrapolated the molar ratio between G3BP1 and USP10 from our input CE proteomic dataset (iBAQ values).

G3BP1 mean iBAQ values	20.24
USP10 mean iBAQ values	14.89
G3BP1/USP10 (Log2)	5.35
G3BP1/USP10 (absolute ratio)	40.85

G3BP1 is present at approximately a 40-fold molar excess compared to USP10 in U2OS CEs. Effective inhibition of SG-like condensate formation in CEODs required at least a 10-fold molar excess of USP10 peptide relative to YFP-G3BP1 to competitively disrupt endogenous interactions and prevent liquid-liquid phase separation.

5. Supplementary Fig9a: Nuclease treatment does seem to have an effect on translation efficiency in Huh7 cells. However, this is noted as non-significant. Please explain.

We thank the reviewer for identifying this error. We have corrected the p-value (now Fig. 6c).

6. Stress granule induction assays often involve oxidative stress. Can authors note if oxidative stress works to induce stress granules in CEODs?

We thank the reviewer for this valuable question. We initially attempted to trigger G3BP1 LLPS in CEODs using sodium arsenite, a well-established inducer of oxidative stress. However, results were somehow intriguing. While arsenite repressed translation by at least 50% at high concentrations, it did not induce YFP-G3BP1 condensates in CEODs. We figured out that arsenite induces pleiotropic toxicity, which directly targeted the luciferase activity and likely many more enzymes. This outcome evidenced the unsuitability of arsenite as a stress inducer for the CEOD approach. These results are now included in the manuscript as a resource for the community. They are described in the Discussion section (page 10, lines 424-432) and shown in new Supplementary Fig. 18.

Reviewer #1 (Remarks to the Author):

This revision has predominantly satisfied my previous concerns.

In the following sentence (line 684-686): “Fifteen μ l of translation reaction mix were layered on a mixture of fluorinated oil (Novec 7500, >99%, 3M) and a PEG-based fluorosurfactant (O08-FluoroSurfactant-1G, Ran Biotechnologies)”, the ratio of fluorinated oil to PEG-based fluorosurfactant in the mixture should be stated.

We apologize for the lack of clarity. The translation reaction is mixed with a fluorinated oil mixture containing 1.4% PEG-based fluorosurfactant at a 1:3 (v/v) ratio. We have added this information in the revised manuscript (lines 690-693).

With regards to the identification of proteins by mass spectrometry, I do share reviewer 2’s concerns. While this approach favors the inclusion of “shell” proteins identified in their purified mass spectrometry sample, it does so at the cost of increased false positives (increased SG proteome) found in the Mock samples. The author’s do address these issues in the discussion section, but perhaps greater attention could be given to this issue. As the authors’ state in their rebuttal letter, there is no unbiased method for the purification of SG, so any method has drawbacks. Regarding the inclusion of mitochondrial proteins in the dsRNA sample, this could be due to inclusion of these proteins in the condensate or a false positive due to the inclusion of pelleted mitochondria at high speed. Would it be possible to visualize whether these proteins localize within the CEOD condensate following treatment with dsRNA?

We thank the reviewer for their thoughtful comment. While we acknowledge that a pelleting-based approach might introduce false-positive proteins, we want to emphasize again that the mitochondrial proteins identified in the dsRNA-induced condensates are not artifacts of the methodology. Notably, mitochondrial gene ontology terms appear exclusively in the dsRNA-induced condensates and are absent from the mock-, silvestrol-, and PEG-induced condensates. If the pelleting step had caused this “contamination”, these mitochondrial proteins and GO terms would be similarly enriched in all conditions, which is clearly not the case. Of note, although annotated as mainly mitochondrial in the COMPARTMENTS database, several of these proteins are also predicted to localize to the cytosol. Nonetheless, and given that no other dsRNA-induced SG purification dataset is currently available for comparison, we understand and carefully consider the concerns outlined by this reviewer.

Following the reviewer’s suggestion, we have analyzed the localization of four mitochondrial proteins (MGME1, MRRF, NDUF56 and MRPL51) that belong to the dsRNA-specific cluster. We assessed their localization in CEODs within YFP–G3BP1 condensates after dsRNA treatment and compared it with their localization following silvestrol treatment. The results, now presented in Supplementary Fig. 16b-c, show specific colocalization of NDUF56 and MRPL51 with YFP-G3BP1 in dsRNA-induced condensates, and an enrichment of MGME1 and MRRF in dsRNA-induced condensates compared to silvestrol-induced condensates. The extent of recruitment to condensates closely reflects their median-normalized iBAQ intensity values as detected by mass spectrometry. Furthermore, we have shown that in cells, MGME1 specifically colocalizes with dsRNA-induced SGs, but is not recruited to silvestrol- or arsenite-induced SGs, confirming the recruitment of MGME1 to SGs only under specific stress conditions. This additional information has now been incorporated in Fig. 9d and e. Altogether, these results strengthen our previous conclusion that certain mitochondrial proteins are specifically recruited to dsRNA-induced condensates and SGs, underscoring the stress-specific composition of SGs.

Reviewer #2 (Remarks to the Author):

Major

1. As it stands, claiming that over half of the proteome (6,060 proteins) is associated with stress granules diminishes the excitement and value of the CEOD system. The lack of specificity is likely due to the experimental method used—simple centrifugation of cell extracts (CEs)—to isolate condensate-associated proteins.

We respectfully disagree with this criticism, as it appears to stem from a conceptual misinterpretation. We do not claim that all proteins are incorporated into SGs, but rather introduce a comparative and quantitative approach for discovering previously unrecognized SG components. Unlike other “classical” SG purification approaches, which aim to define a single 'ground truth' SG proteome, the CEOD system is designed to capture the compositional states of condensates across different stimuli. The broad initial dataset is therefore a prerequisite for identifying stimulus-specific enrichment patterns, and is not a methodological weakness.

Regarding the supposed lack of specificity, both the original and the first revised manuscript provided evidence that the protein cluster selectively enriched in dsRNA-induced condensates consists largely of low-abundance or input-undetectable proteins. This is in clear contrast to PEG-induced condensates and documents specificity. However, as detailed in our response to Reviewer 1, we have now added further experimental validation showing that several of these proteins indeed localize to YFP-G3BP1 condensates in CEODs (Supplementary Fig. 16b-d). In addition, we demonstrate that MGME1 specifically localizes to dsRNA-induced SGs in U2OS cells, but not to silvestrol- or arsenite-induced SGs (Fig. 9d, e). This targeted validation provides evidence for the effectiveness of our approach and supports the biological relevance of the CEOD-derived enrichment patterns, underscoring the meaningful contribution of this work to advancing understanding of stress granule composition.

I encourage the authors to utilize the quantitative information identified across 4-5 replicates in each condition to determine a very stringent threshold that shows significant changes. Please provide a list of candidates that can be utilized as a resource. The authors can use the same criteria used to narrow down potential candidates that were followed up on. For example, authors can provide a high-confidence list based on the fold-change observed between mock and treatments that induce physiologically relevant stress granules in CEODs (e.g., Silversterol and dsRNA), but leave out PEG, which shows completely different enrichment of transcriptome. Once more stringent proteomics data are presented, I recommend repeating the GO enrichment analysis and comparison to published proteomes.

The reviewer suggests excluding the PEG condition due to its distinct transcriptomic profile, compared to the other stress conditions used in our study. We respectfully disagree with this recommendation. The PEG treatment was intentionally included in our analysis as a non-physiological control precisely to broaden the analytical scope and to explore the diversity of condensate compositions arising from different types of treatments. PEG-induced condensates have unique properties that demonstrate how molecular crowding and condensation, based on volume exclusion, alters composition, enriching highly abundant proteins, in contrast with the selective enrichment of low-abundance proteins observed upon dsRNA and silvestrol treatments. Excluding this condition would therefore weaken, rather than strengthen, the central argument of the paper, and limit our understanding of SG biology more broadly. We thus maintain that including PEG provides valuable insights into both shared and distinct features of

SGs across physiological and non-physiological conditions. This rationale is now explained in more detail in line 344-349.

However, as requested by the reviewer, we have performed an additional and more stringent comparative analysis to further prioritize dsRNA- or silvestrol-selective hits. To address the issue of undetectable proteins in input and mock samples, we conducted a differential abundance analysis comparing proteins selectively enriched in dsRNA relative to PEG with those enriched in silvestrol relative to PEG. These proteins were selected using the following cut-off criteria: selective hits must display a fold-enrichment $\text{Log}_2 > 2.5$ when compared to PEG, while not being enriched in the other comparison (fold-enrichment $\text{Log}_2 < 1$) (Rebuttal Fig. R1, left panel and Fig 9b).

Rebuttal Fig. 1. Statistical analysis of cellular proteins selectively enriched in CEODs upon dsRNA or silvestrol treatment. *Left panel:* scatter plot displaying the results of Student’s t-test significance (two-sided, $S_0=1$, p -value <0.05 ; $FDR<0.05$) for dsRNA vs. PEG treatment (y-axis) plotted against silvestrol vs. PEG significant. Proteins with a fold-change of $\text{log}_2>2.5$ displaying minor enrichment in the other comparison ($\text{log}_2<1$) are shown in orange and blue. *Right panel:* Heatmap displaying all 147 selective dsRNA hits from the orange selection on the left. Median-normalized iBAQ intensities across individual replicates and experimental groups are shown.

Consistent with our original ANOVA-based selection (new Fig. 9a and new Supplementary Table 5), this double-pairwise comparison strategy confirmed and identified a unique cluster of 147 cellular proteins that are significantly and reproducibly enriched exclusively upon dsRNA treatment. As shown in the right panel of Rebuttal Fig. R1, this subgroup of proteins was consistently identified across all 4 biological replicates of the dsRNA group and exhibits a highly selective enrichment relative to all the other treatments. Notably, as observed in our previous analysis, the vast majority of these proteins are either absent (not detected) or present only at very low levels in the input samples, indicating that dsRNA-induced condensates selectively enrich low-abundance proteins. The results of this additional analysis, including statistical testing, are now displayed as additional columns in the fully searchable Supplementary Table 5 “Condensate quantitative LC/MS proteomics”. In line with the reviewer’s request to enhance accessibility to potential SG candidates, we now provide this shorter “dsRNA-selective hits” list, including protein names and IDs, in the new Supplementary Table 7 “Selected-hit-lists”. Of note, this new Supplementary Table also displays the “silvestrol-selective hits”, as well as the “SG client list” of 452 proteins we prioritized to identify new client proteins and from which we experimentally validated six novel SG candidates, including FBXO3 (Fig. 10).

The new GO term analysis on this shorter dsRNA-selective hit list, included in new Fig. 9c and new Supplementary Table 6 “GO-term-analysis”, confirms that the biological processes related

to mitochondria genome maintenance and translation are overrepresented, alongside processes involved in cardiolipin biosynthesis process and respiratory chain complex IV assembly. In response to Reviewer 1, we have validated four dsRNA-exclusive candidates from this list (MGME1, MRRF, NDUFS6, and MRPL51). We have demonstrated that MGME1, an exonuclease involved in mitochondrial genome maintenance and also predicted to have cytosolic localization, specifically localizes in cells to dsRNA-induced SGs, but not to silvestrol- or arsenite-induced SGs. These results are now shown in Fig. 9d, e and Supplementary Fig. 16b-c.

We respectfully disagree with the reviewer's assertion that the list of candidates provided in the original and first revised submission could not serve as a resource. While we recognize that prioritized lists may appear more straightforward, our table (now Supplementary Table 5 "Condensate quantitative LC/MS proteomics") was deliberately structured to function as a comprehensive and flexible resource, enabling researchers to efficiently identify, filter, and retrieve at glance proteins according to their specific analytical objectives. It includes multiple alternative keys (a screenshot of the table headers is provided in Rebuttal Fig. 2) based on: (i) the classification of proteins in their respective cluster, (ii) their intersection with all relevant published studies mentioned (Millar *et al.* 2023 for SG core, SG tier 1 and SG and P-bodies tier1, and Freibaum *et al.* 2021 for lysate granules), and (iii) the results of multiple statistical testing, to additionally shortlist candidates based on the specific p-values (ANOVA for unsupervised hierarchical clustering) and the relative fold-enrichment (Student's t-test for pairwise analysis). Altogether, we believe we provide a user-friendly, fully searchable resource that facilitates candidate protein selection through rigorous statistical testing and direct comparison with current literature.

Protein IDs	Majority protein IDs	Protein names	Gene names	Mock only proteins in SGs	In dsRNA cluster	In silvestrol cluster	In dsRNA+silvestrol cluster	In PEG cluster	ANOVA Significant	-Log ANOVA p-value	ANOVA q-value	dsRNA selective hits (compared to PEG and silvestrol)	Silvestrol selective hits (compared to PEG and dsRNA)	Student's T-test Significant dsRNA_PEG	Student's T-test Significant Silv_PEG	Student's T-test Significant	-Log Student's T-test p-value dsRNA_PEG	Student's T-test q-value dsRNA_PEG	Student's T-test Difference dsRNA_PEG	Student's T-test Difference Silv_PEG	
ADAM24RBG1, ADAM24RBG2, Diphosphoheistol NUOT4										4.75085971966	8.233552613096						dsRNA_PEG_Silv_PEG_5_5769028883646	0.00028282828282828	-5.30718994140625	-3.822461443826	
ADAM24RBG1, ADAM24RBG2, Diphosphoheistol NUOT4										1.580151541374	6.266181833905						dsRNA_PEG	0.364462809917366	0.57456253967285	2.484650724248	
ADAM24RBG1, ADAM24RBG2, Diphosphoheistol NUOT4										1.19812686369	0.085857676746						dsRNA_PEG_Silv_PEG_7_1301268505799	0.415089128666317	-0.36840872764874	0.31875053305	
ADAM24RBG1, ADAM24RBG2, Diphosphoheistol NUOT4										7.24917610030	0						dsRNA_PEG_Silv_PEG_P7_3003232808349	7.81022079103208	0.239494779586792	2.643183575667	
ADAM24RBG1, ADAM24RBG2, Diphosphoheistol NUOT4										6.68912096610	0						dsRNA_PEG_Silv_PEG_8_8081753869177	7.78947607294616	0.21844002160176	2.4077884272	
ADAM24RBG1, ADAM24RBG2, Diphosphoheistol NUOT4										0.23501766006	0.3357213014115						dsRNA_PEG_Silv_PEG_9_3486215972181	0.30432008134342	0.8072912082721	0.40265650208	
ADAM24RBG1, ADAM24RBG2, Diphosphoheistol NUOT4										6.5440222006	3.155883893485						dsRNA_PEG_Silv_PEG_5_58474180271892	0.004387665198278	2.028488531312	1.78652501492	
ADAM24RBG1, ADAM24RBG2, Diphosphoheistol NUOT4										3.87932493317	3.4889291258066						dsRNA_PEG_Silv_PEG_3_9764046163031	0.4866310404278	0.54901362459186	1.6390061381	
ADAM24RBG1, ADAM24RBG2, Diphosphoheistol NUOT4										6.58973020574	0						dsRNA_PEG_Silv_PEG_5_9320720267897	0.00024778762161646	4.95893320574036	3.75164170785	
ADAM24RBG1, ADAM24RBG2, Diphosphoheistol NUOT4										2.78022727686	0.0020338483026						dsRNA_PEG	2.8789900443876	0.00371068842328	1.5191293136079	1.1969705423
ADAM24RBG1, ADAM24RBG2, Diphosphoheistol NUOT4										3.47392613362	7.809847198411						dsRNA_PEG	4.5156770130891	0.268836931565996	0.40964646936057	2.86514794246

Rebuttal Fig. 2. Snapshots of the Protein Resource File. Screenshot of the resource file containing detailed information on SG-associated proteins, including individual IBAQ values, enrichment data across stress conditions (dsRNA, silvestrol, PEG), and statistical analysis. This file enables filtering by protein abundance, stress-specific enrichment, and significance, serving as a comprehensive resource for identifying novel SG client proteins.

We believe that this more focused approach adds value to our study and highlights the potential of our proteomic data to uncover new insights into SG biology.

2. Justification of the quality of the proteomics data is poorly done. In Fig. 9C, the authors used this large data set (6,060 proteins) to measure recovery of the known standard stress granule proteome (Tier 1 in RNA granule), stress granule core, and lysate granules. This is an inappropriate comparison, since the size of the starting dataset is so large. The authors should test if this overlap is higher than random using methods like the hypergeometric test. I also

recommend the use of the updated stress granule proteome (407 proteins) reported in Millar et al., 2023.

We respectfully disagree with the reviewer's characterization that condensate proteomes are of poor quality, poorly analyzed, and lack specificity due to the large number of identified proteins. This criticism overlooks the conceptual foundation of our approach. As outlined in the first revision, the strength of our study lies in the systematic and comparative analysis of condensate composition under distinct chemical and biological stress conditions, which fundamentally distinguishes our work from earlier, single-condition studies. Our data consistently demonstrate that the proteins enriched in condensates are strictly determined by the specific stimulus, providing strong evidence for a high level of specificity. In addition, multiple orthogonal validation experiments have confirmed the selective recruitment of several previously unrecognized proteins to SGs in intact cells (new Fig. 9, 10, and Supplementary Fig. 14-19), validating the usefulness of the CEOD approach.

Our initial selection of 280 high-confidence SG-specific proteins (SG tier 1) was deliberate, as this subset provides the most reliable and biologically meaningful representation of SG-associated components. In the proposed updated SG proteome, the remaining 127 proteins are annotated as both SG- and P-body-associated proteins and were thus intentionally excluded to maintain specificity in our SG-focused analysis. Nevertheless, as requested by this reviewer, we have now also included this dataset for comparison and performed the required additional statistical testing, namely by applying a Fisher's exact test to the multiple lists requested (new Supplementary Fig. 14b). Specifically, we applied a Benjamini-Hochberg FDR corrected Fisher's exact test (FDR= 0.02) to determine if there are non-random associations between a categorical column and all terms in the other categorical column. This test uses the hypergeometric distribution to find the exact probability of observing data in a contingency table, by calculating the likelihood of all possible tables with the same marginal totals, assuming no association under the null hypothesis. These results, now presented in new Supplementary Fig. 14b, show that proteins of the SG tier 1, SG-P-bodies tier 1, SG core lists display a significant and non-random enrichment in our datasets.

3. In Figure 8d, the authors have used the full transcriptome (14,324 transcripts) identified instead of the stress granule-enriched transcriptome (1,626) from Khong et al., 2017. This comparison to the full reported transcriptome is inappropriate. Please compare to the stress granule-enriched transcriptome as reported in Khong et al., not the entire transcriptome. Similar to my previous comment, given the large number of transcripts identified, please use statistical methods like the hypergeometric test to show significance.

We thank the reviewer for this important comment. In the revised manuscript, new Fig. 8c presents a comparison between the condensate transcriptomes and the arsenite-induced SG-enriched transcriptome (1,626 mRNAs) reported by Khong *et al.*, 2017. To address the reviewer's request for statistical rigor, we also performed hypergeometric enrichment tests for each of our four condensate conditions (Figure 8c). With the expected exception of PEG, we observed a significant overlap (>38%) for each condition with the published arsenite SG transcriptome, demonstrating that our condensate-enriched mRNAs largely correspond to canonical SG-associated transcripts while also identifying additional, condition-specific RNAs that likely reflect the distinct stressors used in our study.

4. The authors report that the CEOD transcriptome and proteome of the mock condition were already enriched for known components, suggesting pre-existing stress granule seeds. An

alternative explanation is that the mock CEODs activated the integrated stress response during CE isolation using hypotonic stress. In the rebuttal figure 2, mock CE and CEODs show an elevated p-eIF2 α level, indicating moderate activation of the integrated stress response. This figure also shows activation of p-p38 in mock-treated CE CEODs. Given that the main purpose of this manuscript is to report a new CEOD system for studying stress granules, these data should be included in the manuscript as part of CEOD characterization.

We have now incorporated the information on the activation of the ISR as part of the CEOD characterization in the revised manuscript (Supplementary Fig. 1g, h, lines 148-149). Our previous work involving quantitative mathematical modeling of the ISR (Klein *et al.*, 2022) demonstrated that ISR activation requires exceeding a defined threshold—approximately 30-40% eIF2 α phosphorylation, depending on the stressor—before SG formation occurs in cells. Nonetheless, we agree with the reviewer that, while the interpretation involving pre-existing SG seeds is consistent with current models of SG dynamics, the potential contribution of partial eIF2 α phosphorylation cannot be completely excluded. Accordingly, we have removed the section discussing pre-existing seeds from the revised manuscript.

Minor: Supplementary Figure 13b shows a comparison of the total proteome vs. the CEOD proteome. While most of the indicated stress granule proteins (e.g., G3BP1, CAPRIN and novel stress granule proteins) are similar, USP10 is significantly less in CEOD. This is worth mentioning in the main manuscript, since USP10 is a negative regulator of stress granule formation (Kedersha *et al.*, JBC 2016, and Sanders *et al.*, Cell 2020).

We thank the reviewer for this comment. We have now added a statement to the main text (lines 327-329) indicating that the expression levels of G3BP1 and Caprin1 in CEs, i.e. our U2OS cells, are comparable to those reported for the published U2OS cell proteome, whereas USP10 levels are slightly reduced, reflecting cell line-specific variations. Furthermore, we expanded the discussion (lines 442–446) to underscore that variations in the expression levels of SG regulators, such as Caprin1 and USP10, as well as differences in the overall molecular composition of cell lines, should be carefully considered, as they may influence LLPS behavior.

Response to Referees

Reviewer #1 (Remarks to the Author):

The authors have addressed all of my concerns through additional experiments and/or text edits. The data showing MGME1 in cellular SGs is especially strong.

We thank the reviewer for recognizing the additional work and newly presented results.

Reviewer #2 (Remarks to the Author):

The authors have added new data and analysis of the transcriptome and proteome data in the revised manuscript, which did improve on some areas. However, the authors have not appropriately addressed my concern about presenting 6K proteins as ‘proteins associated with the condensates’.

In the revised manuscript, the authors still interpret the 6K proteins identified across all conditions as candidates for the stress granule proteome, as stated in the results section: “The quantitative proteomic analysis of condensates identified a remarkably high number of proteins associated with the condensates (Supplementary Fig. 14a, Supplementary Table 5).”

Currently, the human proteome detectable using highly sensitive mass spectrometers ranges between 7~8K (timsTOF pro) to 10K (Thermo Astral). Interpreting 60% of the detectable proteome as the potential condensate proteome is not scientifically sound. To provide perspective, the 5~6K proteins that the authors propose as potential condensate proteome are 10~20x higher than those reported by other published methods. Furthermore, the lack of selectivity and specificity is evident in their enrichment analysis; dsRNA and silvestrol-induced proteome shows negligible enrichment factors, ranging from 1.08 to 1.14 (Supplementary Figure 14b).

The authors argue that the quantitative analysis provided in Supplementary Table 5 should be mined as a resource for the community. I reviewed the table again. Limiting the proteins that are significantly different between physiologically relevant stressors (dsRNA or silverstrol) and the crowding agent, PEG—using filters provided for the Column 25 (ANOVA significant) and Column 35 (Student’s T-test significant)—still revealed a large number, 5,086 proteins. This number even exceeds the estimated number of RNA-binding proteins in the human proteome (~4K).

I disagree with the authors’ claim that the dataset provides a useful resource for the condensate community.

We note that this reviewer’s concern arises from considering the full dataset (~6,000 proteins) without taking into account the additional filtering steps that were already included in the previous versions of the manuscript. We did not interpret the ~6,000 detected proteins as a “defined stress granule proteome”, but rather as a comprehensive dataset generated under condensate-enriching conditions, which requires downstream prioritization. Importantly, this prioritization was already performed and described in the manuscript:

1. In the first version of the manuscript, we used unsupervised hierarchical clustering as described in Figure 9b and Supplementary Data 3, which identified a total of 315 proteins in the dsRNA cluster (column F of Supplementary Data 3), 755 proteins in the

silvestrol cluster (column G), and 2049 proteins in the dsRNA+silvestrol (column H). This is addressed in lines 344-347 of the manuscript.

2. As requested by this reviewer in the second round of revision, we additionally provided a trimmed list of specific candidates using more stringent filtering by comparing dsRNA- or silvestrol-induced condensates with PEG-induced condensates that reflected non-specific enrichment ($\log_2 \text{FC} > 2.5$). This differential abundance analysis resulted in 147 dsRNA selective hits (column M of Supplementary Data 3) and 14 silvestrol selective hits (column N of Supplementary Data 3). Lists of proteins have also been provided as individual lists in Supplementary Data 5 for simplified retrieval. This is addressed in lines 358-362 of the manuscript.
3. We additionally provided the list of potential SG client proteins we prioritized to identify new SG components based on (i) their absence from the RNA Granule Database, (ii) their predicted cytosolic localization, (iii) antibody availability, and (iv) presence of at least 2 proteins from the same family. This list included 452 proteins (Supplementary Data 5), from which we identified six new SG components which localization was validated in cells treated with dsRNA, silvestrol and arsenite (Figure 10, Supplementary Figures 17-19).

These analyses substantially reduced the dataset to biologically meaningful and more specific subsets, and were already included to address precisely the concern of specificity.

Even if the reviewer disputes the dataset's value as a resource, we shared it transparently with the scientific community. We deleted the sentence line 413 "could serve as a valuable resource for future research".

To avoid any misinterpretation of our statement, we replaced the "quantitative proteomic analysis identified a remarkably high number of proteins associated with the condensates" with "quantitative proteomic analysis of pelleted condensates detected a remarkably high number of proteins."

We also clarified the importance of the hit prioritization in the discussion (line 463-470): "The CEOD approach has several technical limitations. For instance, CE preparation may pre-assemble mRNP seeds; pelleting could co-enrich cellular debris with similar sedimentation properties, induce condensate solidification, inflating false positives (as suggested by the large size of the detected proteins); and washing steps may lose weakly interacting SG clients lacking strong RNA or granule associations. To address potential false positives and identify condition-specific differences from the initial protein lists, it was essential to further prioritize hits using (i) unsupervised hierarchical clustering across all conditions and (ii) differential abundance analysis. Together, these analyses trimmed the lists of specific proteins and revealed clear, biologically relevant differences in condensate composition, distinguishing translation-repression condensates (dsRNA, silvestrol) from non-physiological molecular crowding (PEG)."